

# Long-term Variability and Trends of Agulhas Leakage and its Impacts on the Global Overturning

Hendrik Großelindemann[1,3], Frederic S. Castruccio[2], Gokhan Danabasoglu[2], and Arne Biastoch[1,3]

[1]GEOMAR Helmholtz Centre for Ocean Research Kiel, Kiel, Germany
[2]US National Science Foundation National Center for Atmospheric Research, Boulder, CO, USA
[3]Christian-Albrechts Universität zu Kiel, Kiel, Germany

**Correspondence:** Hendrik Großelindemann (hgrosselindemann@gmail.com)

**Abstract.** Agulhas leakage transports warm and salty Indian Ocean waters into the Atlantic Ocean and as such is an important component of the global ocean circulation. These waters are part of the upper limb of the Atlantic Meridional Overturning Circulation (AMOC), and Agulhas Leakage variability has been linked to AMOC variability. Agulhas Leakage is expected to increase under a warming climate due to a southward shift in the South Hemisphere westerlies, which could further influence the AMOC dynamics. This study uses a set of high-resolution pre-industrial control and historical and transient simulations with the Community Earth System Model (CESM) with a nominal horizontal resolution of $0.1°$ for the ocean and sea-ice and $0.25°$ for the atmosphere and land. At these resolutions, the model represents the necessary scales to investigate the Agulhas Leakage transport variability and its relation to the AMOC. The simulated Agulhas Leakage transport of $19.7 \pm 3\,Sv$ lies well within the observed range of $21.3 \pm 4.7\,Sv$. A positive correlation between the Agulhas Current and the Agulhas Leakage is shown, meaning that an increase of the Agulhas Current transport leads to an increase in Agulhas Leakage. The Agulhas Leakage impacts the strength of the AMOC through Rossby wave dynamics that alter the cross-basin geostrophic balance with a time-lag of 2-3 years. Furthermore, the salt flux associated with the Agulhas Leakage influences AMOC dynamics through the salt-advection feedback by reducing the AMOC's freshwater transport at 34°S. The Agulhas Leakage transport indeed increases under a warming climate due to strengthened and southward shifting winds. In contrast, the Agulhas Current transport decreases, both due to a decrease in the Indonesian Throughflow as well as the strength of the wind-driven subtropical gyre. The increase in Agulhas Leakage is accompanied by a higher salt flux into the Atlantic Ocean, which suggests a destabilisation of the AMOC by salt-advection-feedback.

## 1 Introduction

Agulhas leakage is part of a current system around South Africa which connects the Atlantic Ocean to the Indian Ocean and is an important component for the global ocean circulation. Warm water from the Indian Ocean flows southward along the coast of South Africa as a western boundary current, the Agulhas Current, until it detaches from the coast and reaches a region of strong westerly winds (Figure 1). The dynamics of the flow then become an interplay between its southward inertia and the wind forcing (Beal et al., 2011): The winds push most of the water back eastwards into the Indian Ocean which is called the Agulhas Retroflection. However, some part of the water leaks into the Atlantic Ocean due to instabilities and non-linear





dynamics. Prominent mesoscale eddies form, the so-called Agulhas Rings, and propagate north-westwards carrying the warm and salty waters of the Indian Ocean into the South Atlantic. This transport of heat and salt into the upper Atlantic Ocean plays a role in the Atlantic Meridional Overturning Circulation (AMOC) and subsequently the global conveyor belt (Biastoch et al., 2008).

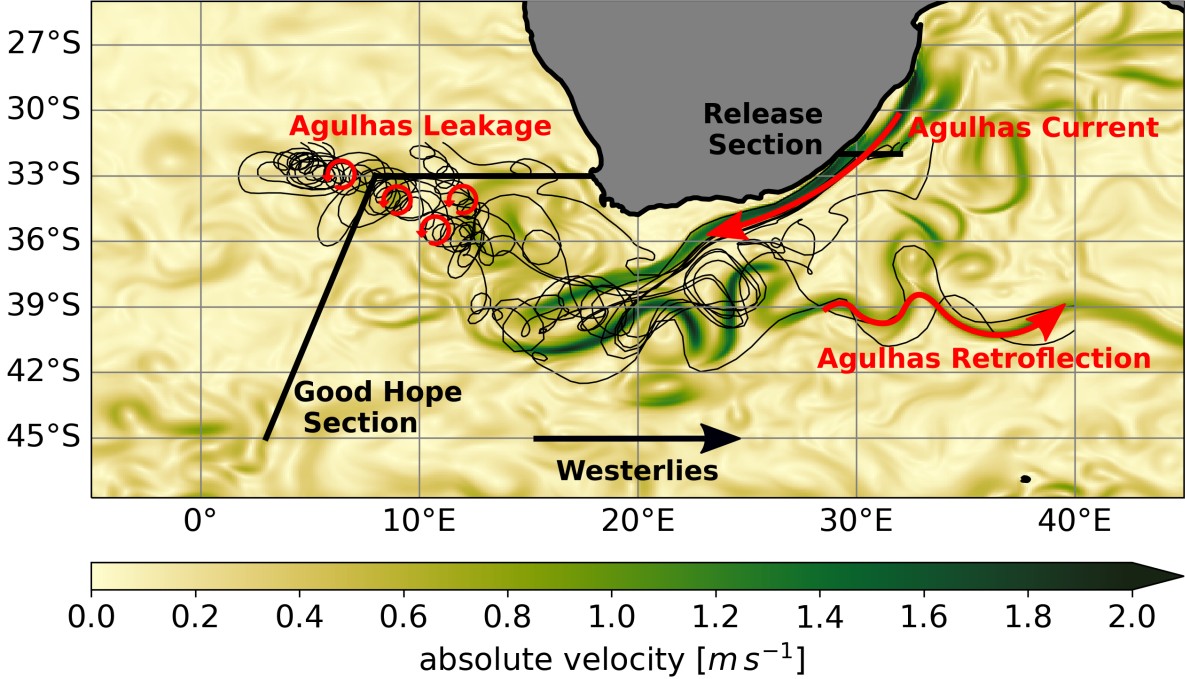

**Figure 1.** Map of the Agulhas region; colours show a snapshot of the five-daily mean velocity magnitude field from the FOSI simulation, thin black lines are example tracks of Lagrangian particles, thick black lines show the release and crossing section and red arrows schematically represent the circulation features.

The trade winds in the Indian and Atlantic Ocean together with the strong westerlies in the Southern Hemisphere form sub-
30 tropical gyres in each basin. Because the African continent does not cross the mean position of the zero line of the wind stress curl, both systems are connected and can be described as one supergyre (Speich et al., 2007). The Agulhas system shapes this connection and therefore opens up a pathway between both ocean basins. The amount of trans-basin transport is controlled by the wind field (Beal et al., 2011). The winds over the Indian ocean determine the inertia of the Agulhas Current, which together with the strength of the westerlies, controls whether the current turns west into the Atlantic or loops back into the Indian Ocean.
Ultimately, it is this interplay between the two that controls how much water is leaking into the Atlantic (Durgadoo et al., 2013). Rühs et al. (2022) found that an increase in the wind stress curl over the Indian Ocean resulting from strengthened winds leads to an increase in the Agulhas Leakage.



Agulhas Leakage is not a straight current that flows into the Atlantic Ocean, but rather consists of mesoscale eddies, which form in the retroflection region and then propagate into the Atlantic (Olson and Evans, 1986). These eddies are affected by the bathymetry and take different routes into the basin, some even reaching the Brazilian coast (Dencausse et al., 2010). The formation rates and propagation speeds of these eddies lead to variability in the overall transport on interdecadal timescales (Holton et al., 2017). The eddy activity is additionally impacted by large scale climate modes such as the Southern Annular Mode or the El-Nino Southern Oscillation (Morrow et al., 2010). Another but not as important pathway for Indian Ocean waters is the Good Hope Jet, described as an extension of the Agulhas Current along the continental slope (Gordon et al., 1995).

Warm and salty surface waters from the Indian Ocean are transported into the Atlantic by Agulhas Leakage, and connect to the upper part of the AMOC as part of the "warm-water route" of the global overturning circulation in the Atlantic Ocean (Gordon, 1986; Rühs et al., 2019). It is hypothesized that these water properties influence the formation of deep water in the North Atlantic mainly through the salt input. Weijer and van Sebille (2014) investigated the influence of salinity input of Agulhas Leakage into the Atlantic on the AMOC on interdecadal timescales. While they could show an advective connection from Agulhas Leakage to the North Atlantic, no impact to the AMOC strength could be detected. However, due to a bias in salinity in the model, the salinity anomalies induced by the Agulhas Leakage were much weaker than in the observations. Biastoch et al. (2008) showed the influence of the Agulhas Leakage, in this case through Rossby and topographic shelf waves, on AMOC variability on decadal timescales is on the same order as that of deep water formation in the North Atlantic. Another study by Biastoch et al. (2015) found that the Atlantic multi-decadal variability (AMV), a basin-wide North Atlantic sea surface temperature fluctuation on multi-decadal time scales, covaries with Agulhas Leakage. Still, the exact relationship between the Agulhas Leakage and the AMOC is not that clear, both for the volume transport as well as for the hydrographic influences.

The stability of the AMOC and hence the possibility of a future collapse is a major topic in current research (Boulton et al., 2014; Hu et al., 2021; Boers, 2021). One theory involved here is the salt-advection feedback. It describes the AMOC stability as a feedback loop between the AMOC strength in the North Atlantic which controls the freshwater transport through 34°S, which influences the density difference between the North and the South Atlantic which then again influences the AMOC strength. The direction of the freshwater transport, either southward or northward, determines the sign of the feedback loop being positive or negative, respectively. More precisely, a negative feedback loop describes a self-stabilising system; an AMOC change is balanced by the other components which then revert the change back to its original state. A positive loop however describes the opposite; an AMOC weakening f.e. is enhanced by the other components which further increases the weakening and leads to an unstable system. The process originates from a theory by Stommel (1961) based on simple box models and was further developed by Rahmstorf (1996). Only a negative freshwater transport allows for a bi-stable AMOC regime as is suggested for the real ocean, where a sudden shift in the freshwater forcing can lead to an AMOC collapse (Rahmstorf, 1996). The impact of the Agulhas Leakage on this freshwater transport and further impacts on the AMOC remain to be completely understood (Weijer et al., 2019).

Furthermore, the Indian Ocean Throughflow (ITF), a connection between the Pacific and the Indian Ocean, has an impact on Agulhas Leakage. Le Bars et al. (2013) and Makarim et al. (2019) showed that the ITF influences the strength of the Agulhas Current and subsequently Agulhas Leakage. Van Sebille et al. (2009) have shown that an increase in the Agulhas



Current strength leads to a decrease in Agulhas Leakage due to higher inertia of the Agulhas Current and therefore stronger retroflection. However, this result has been under discussion since then and Loveday et al. (2014) describe a decoupling of the

Agulhas Leakage strength from Agulhas Current variability. Zhang et al. (2023) found variability in the Agulhas Current on decadal and multi-decadal timescales, which might impact Agulhas Leakage depending on the exact relation between them. This highlights the complexity of the region and how many factors can play a role in determining the strength of Agulhas Leakage.

It is projected and already observed, that the Southern Hemisphere westerlies are strengthening and moving southward under

climate change (Cai, 2006). This has direct impacts on the controlling dynamics of the Agulhas Leakage. Biastoch et al. (2009) related the southward shift to an increase in leakage and by that a salinification of the South Atlantic. A modelling study by Beech et al. (2022) shows an increase in eddy activity in the region and a connected increase in Agulhas Leakage by up to 6 Sv due to climate change. Additionally, Ivanciu et al. (2022) used a high-resolution nesting approach as well as interactive ozone forcing to show that Agulhas Leakage is increasing by 1.5 Sv over 80 years. A recent study by Li et al. (2022) highlights the

importance of a poleward shift of mid-latitude easterlies globally in controlling the southern boundary of subtropical ocean gyres and subsequently the southward extent of western boundary currents.

Due to the fact that mesoscale dynamics are highly important in the Agulhas region and especially in Agulhas Leakage, these processes need to be resolved to capture the Agulhas Leakage transport (Schubert et al., 2021). Over the past decade, growth in ocean modelling capabilities allowed for eddy-resolving resolutions, i.e. grid sizes smaller than the local Rossby

deformation radius, in the Agulhas region and therefore its investigation. However, these were mainly hindcast simulations covering around 60 years from 1960 to 2020, which limits the modes of variability that one is able to extract. Little is known about the variability of the system on timescales of decades to centuries (Beal et al., 2011; Rühs et al., 2022). In order to investigate this, high-resolution and multi-centennial model configurations are needed. The model used for this study is a high-resolution configuration of the Community Earth System Model (CESM) which consists of a 0.1° ocean and therefore

resolves mesoscale processes (Chang et al., 2020). The global high-resolution is particularly important to better represent the global overturning circulation (Roberts et al., 2020). The available CESM simulations include a 500-year pre-industrial control (PIcontrol) run and future projections under different scenarios following the Representative Concentration Pathways (RCP) (Van Vuuren et al., 2011). These offer a unique opportunity to examine Agulhas Leakage variability on timescales of decades and longer, allowing an assessment of possible changes in Agulhas Leakage in a warming climate. Additionally, one

can investigate the connection of Agulhas Leakage to the global climate in a self-consistent, fully coupled Earth System Model as opposed to forced regional ocean-only simulations.

This study is structured as follows: We first describe the model and simulations, then explain how we estimate Agulhas Leakage transport with a particle tracking algorithm and the applied analysis methods. We then investigate the following questions: What is the internal variability of Agulhas Leakage on decadal to multi-decadal timescales and what are related

driving mechanisms? How is Agulhas Leakage connected to the global overturning circulation and specifically the AMOC? And how might the system change in a warming climate? We end with discussing the results with existing literature and concluding our findings.





## 2 Model and simulations

This study uses the high-resolution version of the Community Earth System Model (CESM-HR; Chang et al. (2020)). CESM-
HR is based on an earlier model version, CESM1.3, as described in Meehl et al. (2019) with many additional modifications and
improvements over its standard version. The ocean component is the Parallel Ocean Program version 2 (POP2 ; Danabasoglu
et al. (2012); Smith et al. (2010)) with a 0.1° horizontal mesh and 62 vertical levels of increasing layer thickness to the bottom.
The atmosphere component is the Community Atmosphere Model version 5 (CAM5 ; Neale et al. (2012)) and has a horizontal
resolution of 0.25° and 30 vertical levels. The other components are the Community Ice Code version 4 (CICE4; Hunke and
Lipscomb (2008)) and the Community Land Model version 4 (CLM4 ; Lawrence et al. (2011)) with resolutions of 0.1° and
0.25°, respectively. With this configuration, a 520-year pre-industrial control run was performed. In our study, we use model
years 150 to 520, avoiding the initial spin up period. Figure 2 shows all used configurations, their time periods and forcing
types.

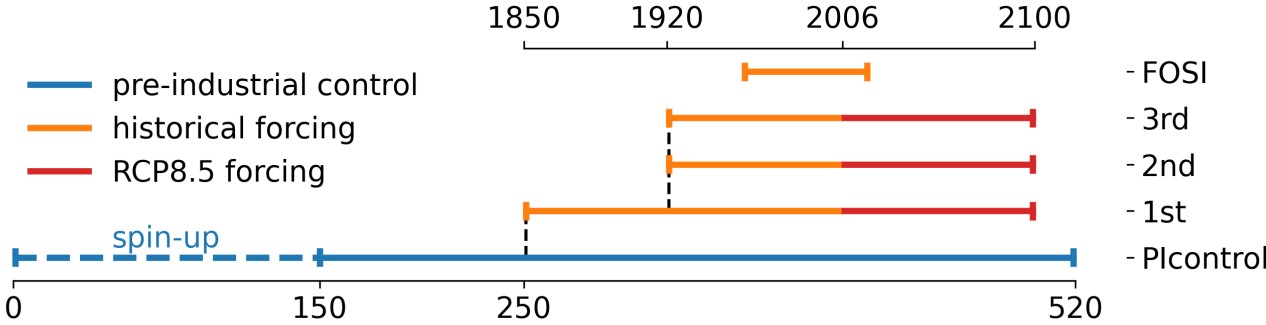

**Figure 2.** Schematic of the CESM simulations used in the manuscript; the PIcontrol and the three ensemble members use the fully-coupled
CESM while FOSI represents a Forced Ocean - Sea Ice simulation using the same CESM ocean and sea-ice models. Colours indicate different
$CO_2$-forcing types and the black dashed lines show branching of another run.

It has been shown that this simulation captures many features of the current climate well, showing improvements compared
to a lower resolution counterpart (Chang et al., 2020). With the same configuration, three ensemble runs were performed
applying a transient $CO_2$-forcing to simulate climate change. The first member was branched from PIcontrol at model year
250 or calendar year 1850. The second and third members were branched from the first member at year 1920, because the first
70 years contain little forcing, saving computing time. Historically observed $CO_2$ forcing was applied until the year 2006 and
then the RCP8.5 scenario (Van Vuuren et al., 2011) was used until 2100. Additionally, we use output from a forced ocean-sea
ice (FOSI) hindcast simulation for the 1958-2018 period. FOSI employs the same ocean and sea-ice components, forced with
the JRA55-do dataset from Tsujino et al. (2018).

We use satellite based absolute dynamic topography (Sea Surface Height above geoid) daily means from 1993 to 2018 to
evaluate the model's capability to resolve the necessary mesoscale activity in the region and the overall large scale circulation



pattern. This product is provided by the Copernicus Marine Environment Monitoring Service (CMEMS) on a global grid of
0.25° resolution and is based on a data unification and altimeter combination system (Mertz et al., 2017).

## 3    Methods

### 3.1    Agulhas leakage Estimation

To calculate annual Agulhas leakage transport, we follow a well-established methodology (Schmidt et al., 2021; Rühs et al.,
2022). We perform a Lagrangian particle tracking method, utilising the python package Oceanparcels (Delandmeter and Van
Sebille, 2019; Kehl et al., 2023). Particles are released every month of the model output along a section at 32°S from 29-32°E
through which the Agulhas Current flows southward along the coast of South Africa (Figure 1). An initial transport is assigned
to each particle, based on the volume around them and the velocity at that point and then tracked for five years. The tracking
is an interpolation that applies a fourth-order Runge-Kutta advection scheme (Delandmeter and Van Sebille, 2019). We use an
internal timestep of 32 minutes for the advection scheme, based on the grid size of the model. This timestep keeps the particles
in the same grid box for at least two timesteps with still reasonable computational effort. The annual total Agulhas Leakage
transport then consists of the volume of all particles that cross the Good Hope section as their first exit from the domain during
a five year tracking period. Observations have been done along the Good Hope section since 2004, including conductivity-
temperature-depth profiles and mooring deployments, and it is part of the South Atlantic observing network of the meridional
overturning circulation (Speich et al., 2023). Because the majority of particles cross the section in the first year (Van Sebille
et al., 2018), the whole transport of five years is assigned to the release year. Ideally, this tracking is performed on the full 3D
velocity field and on a high, daily to 5-daily, temporal output of an ocean model. However, the 3D fields were only available
in monthly resolution for the coupled simulations. We therefore tested different techniques using different fields and temporal
resolutions to find the best possible estimate we could get from the PIcontrol. Figure A1 shows decadally-filtered timeseries
(see Section 3.2 for filtering method) of all used approaches for Agulhas Leakage calculation. This effort includes 3D tracking
on five-daily and monthly velocity fields, as well as a regression method based on horizontal geostrophic velocities calculated
from the Sea Surface Height (SSH). This regression method was developed and validated by Rühs et al. (2022). We tested the
regression in the FOSI simulations, where the five-daily 3D velocity field for the 1983-2015 period was available. This FOSI
analysis gives us an estimate of Agulhas Leakage at high temporal resolution. The regression between the 3D (red line) and
the geostrophic surface tracking and then applied to reconstruct the total Agulhas Leakage transport from the surface tracking
(purple line) leads to good alignment between both with a significant correlation of 0.92 based on a 95% level, exceeding a
correlation of 0.86 from Rühs et al. (2022). This indicates that this regression is a valid technique of reconstructing the Agulhas
Leakage transport, when only surface velocities are available.

    In order to find the longest possible, but also plausible timeseries of Agulhas Leakage in the PIcontrol, we compare the
estimated transport from the available fields based on their correlation and variances (Figure A1). Plausibility is however not
easy to determine since differences can be based on true signals in the circulation we want to sample, but also on errors from
the particle tracking e.g. due to lower temporal resolution. We had 370 years of monthly 3D velocity and SSH data from year



to 520 and five daily SSH data from year 338 to 512. Therefore, we had to decide between completely relying on the fully-coupled system or combining the SSH tracking with the regression from the more precise FOSI simulation to get a full transport estimate. We applied the regression on the 174 years of five-daily SSH tracking as our best estimate for high temporal
resolution in the PIcontrol (blue line). Then, we compared this timeseries to the 370 years of monthly data, both for the full 3D tracking (green line) and the SSH tracking with the regression applied (yellow line). We found significant correlations of 0.77 between the five daily SSH and the monthly 3D tracking, while it was only 0.59 between five daily and monthly SSH. On decadally-filtered timeseries, we found a significant correlation of 0.82 between the five-daily SSH and the monthly 3D. For the monthly SSH tracking, the correlation was less with 0.69 and just not significant with a p-value of 0.06. Regarding the
interannual variation of the transport expressed as a standard deviation, a value of 2.48 Sv from the five-daily 3D tracking of FOSI is our best estimate, i.e., *truth*. In PIcontrol, the standard deviations are 2.85 Sv and 3.01 Sv for the monthly SSH and 3D tracking, respectively, which are both larger than, but close to the true value.

We decided to use the monthly 3D tracking method to obtain a 370-year timeseries of the Agulhas Leakage transport because of better correlations and standard deviations of interanually- and decadally-filtered data with the truth than those of the SSH
approach. However, the volume transport from the particle tracking can vary depending on the release section. Schmidt et al. (2021) show an increase of $2\,Sv$ from a release at $32°$ S (as we do) to a release at the ACT (Agulhas Current Time-Series) section at $34°$ S, where the observational estimates from Beal et al. (2015) and Daher et al. (2020) are based on. On the other hand, using monthly data instead of a higher temporal resolution can lead to a more laminar flow with tracking and an associated increase in the volume transport. We found an increase of around $2\,Sv$ due to this in FOSI (red and brown lines in
Figure A1). Both factors, the different release sections and the bias due to the different temporal resolutions, could even out, but we cannot quantify these impacts. Because our focus is on decadal and longer variability, the calculation based on monthly 3D velocities is an appropriate strategy. This is confirmed by Cheng et al. (2016), who showed that monthly mean outputs are sufficient to investigate variability on time scales longer than seasonal. Additionally, the 3D tracking allows us to track the temperature and salinity along a particle's trajectory, which we can use for further analysis. We define the Agulhas Current
transport as the sum of the transport of all released particles. This is different than just calculating the volume transport through a section, because we are only releasing particles at points that have a southward velocity, e.g., into the domain.

### 3.2 Analysis

To evaluate the representation of the ocean circulation around South Africa in our simulations, we first compare simulated and satellite-based Sea Surface Height variability. We use the first ensemble member and the overlapping period from 1993 to
2018 to be most consistent with the observations and calculate their standard deviations. Additionally, observational transport estimates based on floats and drifters for the Agulhas Leakage (Daher et al., 2020) as well as mooring data for the Agulhas Current (Beal et al., 2015) are compared to the results of the particle tracking.

To investigate variability of Agulhas Leakage on different timescales, we perform a spectral analysis using the annual-mean timeseries. Due to the relatively short length of this timeseries data, meaning only 370 data points, this was not a straightforward
exercise. We employ just a general Fourier Transform and not a wavelet analysis to better resolve the longer frequencies. The



significance of the peaks is calculated following Torrence and Compo (1998). In general, significance in this study always refers to the 95% level. To explore relationships between two timeseries, we perform lead-lag correlation analysis. For these, the data have been filtered using a five-year Hanning window and are detrended. Significance is based on a two-sided student's t-test, that incorporates the autocorrelation of each timeseries. We perform coherence analysis to investigate co-variability between
two timeseries and infer possible driving mechanisms. Because the results of the coherence analysis are quite dependent on the window size for the wavelets, we calculate coherence over a range of sizes with zero padding for the smaller ones to keep the same frequency resolution. Then, we average the coherence distributions of all window sizes to get the dominant frequencies. The confidence levels are estimated following Thompson (1979) and are averaged in the same way.

We extract wind metrics as the maximum and minimum zonal wind stress in the region between 20°E to 110°E and 20°S to
60°S, following Rühs et al. (2022). This then also gives us the latitude of maximum zonal wind stress and thereby the position of the Southern Hemisphere westerlies. Additionally, we calculate the average wind stress curl in the same longitudinal range, but between 35°S and 45°S. The regions can be seen in Figure A5. We also calculate the Sverdrup transport at 32°S as an estimation of the strength of the southern subtropical gyre in the Indian Ocean. The Southern Annular Mode is defined as the first Empirical Orthogonal Function (EOF) mode of the annual-mean sea-level pressure south of 20°S and it is calculated using
the Climate Variability Diagnostics Package package (CVDP) (Phillips et al., 2014; Thompson and Wallace, 2000).

Because salinity of each particle along its trajectory is tracked, it is possible to calculate the salt flux of Agulhas Leakage, $F_S$, into the Atlantic Ocean. This has been calculated following the method from Weijer and van Sebille (2014), where the amount of salt, that a particle brings into the Atlantic, is the difference of its salinity $S_i$ at the Good Hope Section and the long-term salinity mean $\overline{S}$ (at the crossing location along the section and at the same depth) times its transport $V_i$ and then sum
up all particles that cross the Good Hope section:

$$F_S(t) = \Sigma_i V_i(t)(S_i(t) - \overline{S})$$

where $t$ is time. Therefore, by referring to the average salinity at the section, salt flux variations are the advection of local salinity anomalies only. Variations in the volume transport alone would not cause any variance in the salt flux (Weijer and van Sebille, 2014). However, as there is a small salinity drift in PIcontrol, i.e., a freshening over time, we use a linear fit of the
section mean over time instead of just the temporal mean. For the transient simulations, the reference salinity was a constant temporal mean from 1920-2100 to allow for a potential trend in the salt flux that is based on climate change and not on a model drift. Additionally, we only choose particles between $150\,m$ and $1500\,m$ to remove mixed layer influences, being consistent with Weijer and van Sebille (2014).

To investigate the impact of the Agulhas Leakage onto the AMOC, we compute AMOC transports at different latitudes. The
AMOC strength is the maximum value of the AMOC stream function in depth space and deeper than 500 m, excluding the surface cells. Additionally, we investigate a relation between the Agulhas Leakage salt flux and the overall freshwater flux at 34°S. We use the method described in Jüling et al. (2021) to calculate the annual freshwater transport $F_{ov}$:

$$F_{ov} = -\frac{1}{S_0} \int (\int_W^E v^* dx)(\langle S \rangle - S_0)dz$$




where $S_0 = 35\,psu$, $\langle S \rangle$ is the zonal-mean salinity, and $v^* = v - \hat{v}$ with $\hat{v}$ being the section-mean meridional velocity. This

freshwater transport is thought to be an important aspect of the salt-advection feedback contributing to AMOC stability.

The ITF is another important pathway for the global overturning and can be used to investigate its connections and changes (Durgadoo et al., 2017). To estimate the ITF volume transport, we calculate the annual-mean transport through sections covering each strait from the Pacific to the Indian Ocean between Thailand and Australia and sum up the sections. We estimate the wind-driven part of the Agulhas Current as the Sverdrup transport at 32°S in order to separate wind-driven from the overturning

circulation. This transport is calculated following the formula from Sverdrup (1947).

When investigating future changes, we take the ensemble mean from the three ensemble members for their overlapping period from 1920 to 2100. While three ensemble members are not that many, it removes some of the internal variability and therefore increases the robustness of examined trends due to global warming.

## 4 Results

### 4.1 Model representation of the region

To determine the fidelity of our simulations in resolving the necessary dynamics in our region of interest, we compare the Sea Surface Height variability from the PIcontrol simulation to that of satellite observations in Figure 3. The strongest variability in the observations appears around 20°E and 40°S. This is where the Agulhas Current reacts to the overlying wind field and is being pushed back to the East. Upsream of this location, the Agulhas Current is rather stable, leading to a corridor of low

variability. The higher variability around Madagascar is due to mesoscale eddies as well, the eddies then feed into the Agulhas Current. The flow back into the Indian Ocean through the Agulhas retroflection is characterised by a meandering flow, which is visible in the Sea Surface Height between 35°- 40°S. Agulhas Leakage itself consists mostly of large scale eddies, that then propagate into the Atlantic, marked by the increased variability in that direction. In the model, the general picture is very similar, meaning that the model represents these observed characteristics well. However, some differences exist as well: the

variability around 20°E is stronger in the model and there is a more pronounced eddy corridor into the Atlantic Ocean. This is a known model deficiency present in simulations with other air-sea coupled models as well (Ivanciu et al., 2022).

Our 370-year long Agulhas Leakage time series shown in Figure 4a shows a transport of $19.7 \pm 3\,Sv$, which lies well within the most current observational estimate of $21.3 \pm 4.7\,Sv$ based on floats, drifters, and a mooring array from Daher et al. (2020) with the caveat that the simulated and observed estimates have slightly different initialization latitudes which can result in slight

differences in the total transports (Schmidt et al., 2021). The simulated Agulhas Current transport of $72 \pm 4\,Sv$ is also similar to the observed transport of $77 \pm 4.0\,Sv$ from mooring data (Beal et al., 2015). We compute the ratios of Agulhas Leakage transport to the Agulhas Current transport as $27.3 \pm 4\,\%$ for the model and $27.6 \pm 2.5\,\%$ for observations. Additionally and as a metric for the global overturning circulation, we find a mean AMOC transport at 26.5°N of $18.26 \pm 1.08\,Sv$ which is in the range of the RAPID array observations of $17.55 \pm 2.88\,Sv$ (Moat et al., 2023; CMEMS, 2023). Thus, these rather good

agreements with observations give us additional confidence in our model's fidelity.



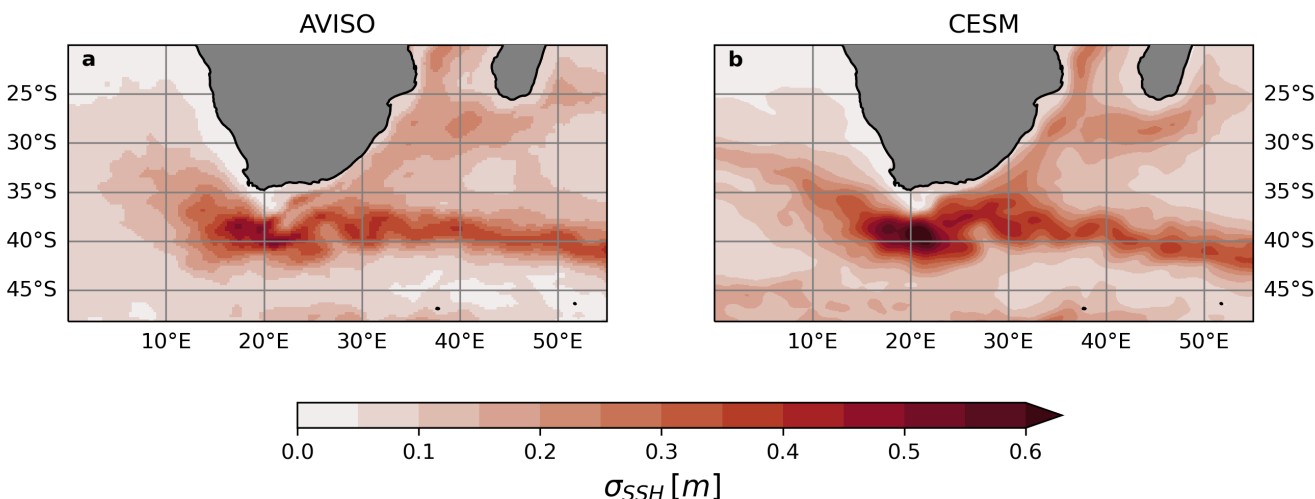

**Figure 3.** Sea Surface Height variability $\sigma_{SSH}$ from daily AVISO satellite altimetry (a) and in the first ensemble member of CESM (b) for the overlapping time period from 1993 to 2018.

## 4.2 Variability on different timescales

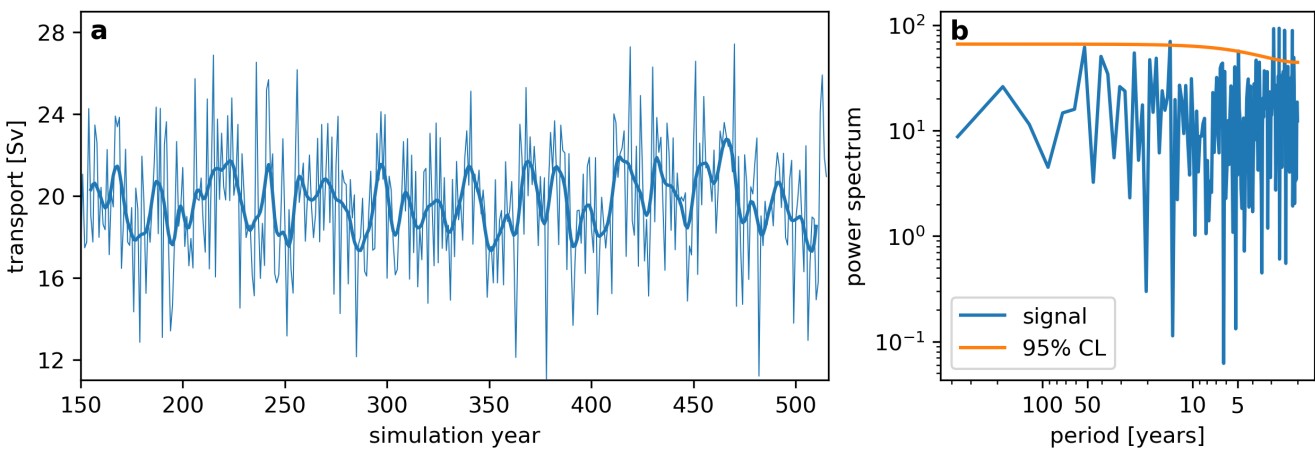

**Figure 4.** Agulhas Leakage timeseries from the Lagrangian particle tracking for PIcontrol (a). The thin and thick lines represent annual-mean and 11-year running-mean timeseries, respectively. The power spectrum of the annual-mean Agulhas Leakage timeseries (b) with the orange line showing the 95% confidence level.

The Agulhas Leakage timeseries from PIcontrol shows a strong interannual variability, while the decadally-filtered timeseries hints at variability on longer timescales. The spectrum of the annual-mean timeseries presented in Figure 4b shows that the interannual variability clearly stands out, which is related to the formation and propagation of Agulhas Rings (Holton et al.,




2017). Another significant peak is seen at a 14-year period and there are peaks also around 40-50 years, but these latter peaks are just below our significance level. To assess the robustness of these peaks, we also performed spectral analysis using different estimates of Agulhas Leakage (not shown). These further analyses indicate that the dominant peaks / frequencies vary considerably and that there is really no robust long-term variability peak that can be identified.

## 4.3   Wind forcing across scales

It is known that Agulhas Leakage is driven by the prevailing wind field over the Agulhas region and the southern Indian Ocean (Beal et al., 2011). We investigate this relationship in the PIcontrol as another proof of concept before looking at longer periods, also exploring the associated timescales.

Performing a lead-lag correlation analysis shown in Figure 5a, we find that both the maximum and minimum zonal wind stress as well as the average wind stress curl over the Indian Ocean show significant lead times of 3-4 years over the Agulhas

Leakage. The correlation values are 0.33, 0.25 and −0.20 for the wind stress curl and the maximum/minimum zonal wind stress, respectively. These are not that high due to the strong interannual variability of the Agulhas Leakage, but still significant because of the length of the timeseries. The lead time of three years relates to a Sverdrup response over the Indian Ocean, that takes some time to propagate into the region (DiNezio et al., 2009). When calculating the correlations for each point in space, presented in Figure A2, we find the strongest correlations around 50°S and 50°E. This co-locates with the maximum long-term

mean zonal windstress, which means that it is indeed a strengthening of the winds that increases the Agulhas Leakage and not a latitudinal shift. Additionally, we find a significant negative correlations at around 30°S and across the Indian Ocean, which is related to strengthened trade winds and hence a stronger subtropical gyre. Barotropic as well as baroclinic adjustment processes can explain the range of lead times of 0-3 years (Anderson and Killworth, 1977).

The coherence analysis shown in Figure 5b shows strong and significant coherence values of up to 0.66 at different periods

including long-term co-variability at 14- and 30-year cycles. The wind metrics, especially the maximum zonal wind stress and the wind stress curl, have very similar coherence spectra, underlining the wind influence and suggesting that the wind is the dominant driver of Agulhas Leakage across timescales.

## 4.4   Connection to local and global circulation

The connection of the Agulhas Current transport to the strength of the Agulhas Leakage has been under discussion in the past.

We calculated the lead-lag relationship between both in the PIcontrol in Figure 6. A positive and significant correlation of 0.19 stands out at zero lag. This means that a stronger Agulhas Current directly leads to a stronger Agulhas Leakage in the same year. The fact that the highest correlation occurs at zero lag is reasonable because we assign the Agulhas Leakage transport to the release year of the particles within the Agulhas Current.

Further into the large-scale circulation of the Atlantic, we find a significant correlation of 0.22 when the Agulhas Leakage

transport leads AMOC at 34°S by 2-3 years. As the majority of the particles reach the Good Hope section at 34°S within one year, this multi-year time lag of the correlation does not fit an advective timescale. However, it can be explained by invoking westward propagating Rossby waves which modify the east-west geostrophic gradient. An indicator of these waves is the depth





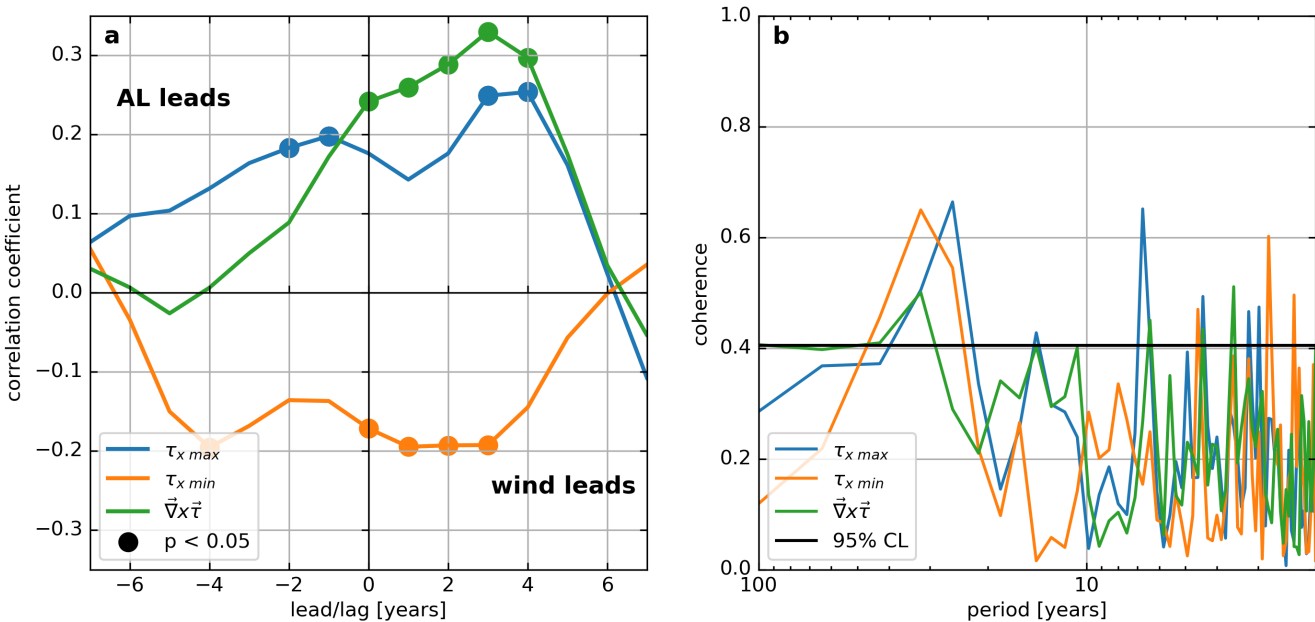

**Figure 5.** Lead-lag correlation (a) and coherence (b) between Agulhas Leakage and the prevailing winds, in particular the maximum/minimum zonal windstress, $\tau_{x\,max/min}$, within 20°-110°E and 20°-60°S and the average windstress curl, $\nabla \times \tau$, over 20°-110°E and 35°-45°S. Significance levels are represented with the dots in (a) and the horizontal line in (b).

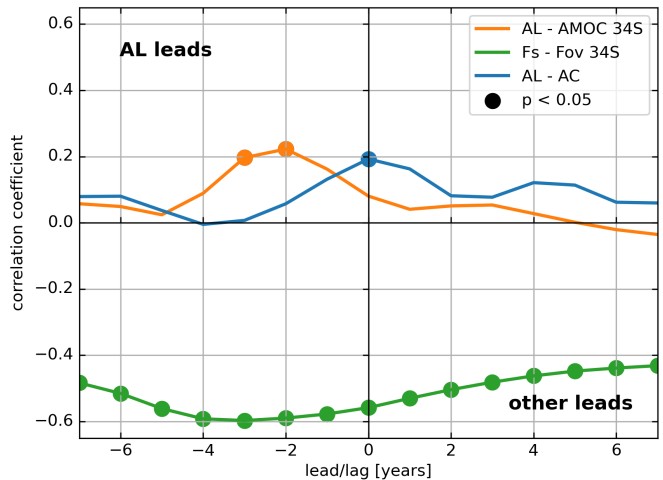

**Figure 6.** Lead-lag correlations between the Agulhas Leakage and the Agulhas Current (AC) and the AMOC strength at 34°S as well as between Agulhas Leakage salt flux ($F_s$) and freshwater transport ($F_{ov}$) at 34°S. The dots denote significance.



anomaly of the 10°C isotherm (Olson and Evans, 1986) as presented in Figure A3. The figure shows a westward propagation
of increased isotherm depth at 34°S within a year (or longer) of strong Agulhas Leakage transport. The signals cross the South
Atlantic basin in around three years. The related wave speed is around $4\,cm/s$ which is in the range of both theoretical and
observation-based wave speed estimates at these latitudes (Osychny and Cornillon, 2004; Webb et al., 2021). However, we
cannot conclusively distinguish between Agulhas Rings and Rossby waves since both would have the same propagation speed.
Overall, the temporal variability of the AMOC cannot be solely linked to Agulhas Leakage but is rather a combination of a
series of different contributing factors including atmospheric forcing and other far-field influences.

Another interesting aspect of the Agulhas Leakage is not just the volume transport itself, but especially the amount of salt
that is brought into the Atlantic Ocean. This salt flux is thought to be the main factor that is impacting the AMOC (Weijer and
van Sebille, 2014) as most of the heat of the Agulhas Leakage waters is lost to the atmosphere in the South Atlantic locally
(Ivanciu et al., 2022). We find a mean northward freshwater transport of $0.1 \pm 0.03\,Sv$ over the time period from simulations
years 150 to 520 which is positive the entire time, meaning a stable salt-advection feedback. The correlation between the salt
flux and the freshwater transport at 34°S (Figure 6) shows a strong negative correlation of $-0.6$ when $F_S$ leads $F_{ov}$ by three
years, similar to the timescale between the Agulhas Leakage and AMOC transports discussed above. This means that a stronger
salt import from the Indian Ocean through Agulhas Leakage reduces the freshwater transport. The salt input of the Agulhas
Leakage is mostly confined to the upper $1000\,m$, which relates to the depth of Agulhas rings (Schmid et al., 2003).

### 4.5 The global overturning circulation under global warming

An important question is how the Agulhas Leakage might change under climate change and what impacts this might have.
Previous studies have suggested that an increase in Agulhas Leakage due to changing wind patterns – an increase in magnitude
and southward shift of the Southern Hemisphere westerlies (Cai, 2006) – over the region has been already happening during the
past decades (Biastoch et al., 2009) and will continue into the future (Ivanciu et al., 2022). Here, we investigate this question
using our three-member ensemble of historical and future transient simulations.

Figure 7 shows the timeseries of Agulhas Leakage from the PIcontrol and of each ensemble member as well as their mean.
We find an increase in Agulhas Leakage of $0.08\,Sv/dec$ during the period from 1920 to 2100 in the ensemble mean. This
trend is significant and also consistent between all three members with only negligibly different magnitudes. It is lower by
more than a factor of 2 from that reported in Ivanciu et al. (2022), i.e., about $0.2\,Sv/dec$. However, we note that the trend is
sensitive to the period used in its calculation. For example, we do not find a discernible trend when calculating it for just the
future forcing period, e.g., from 2005 to 2100. This finding is in contrast with changes in the Southern Hemisphere westerlies
discussed above. To further investigate the changes in the winds, we calculate the Southern Annular Mode over that time period
as a general representation of the wind field. The Southern Annular Mode in Figure 8 shows an increasing and stronger trend
than the Agulhas Leakage. However, an increase in the Southern Annular Mode can result from a variety of changes in the
wind patterns, i.e., either a strengthening or a southward shift and also from regional distributions of these changes. Therefore,
we take a closer look at the wind metrics discussed earlier, considering the decadally-filtered ensemble-mean timeseries from
historical and transient simulations (Figure A4). We find an increase in the maximum zonal windstress over the Indian Ocean





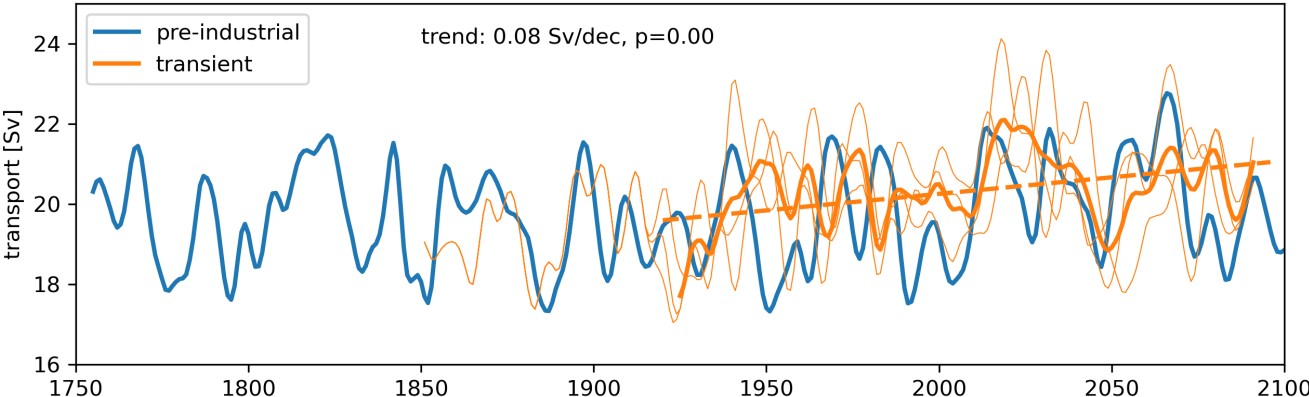

**Figure 7.** Decadally-filtered Agulhas Leakage transport timeseries for PIcontrol, the ensemble mean of three members under historical and RCP8.5 forcing in the thick line, and individual members in the thin lines. Linear trend line and metrics for the period from 1920 to 2100 are also shown.

as well as a southward shift of the latitude of the maximum zonal windstress, both statistically significant. This means that both changes impact the Southern Annular Mode similarly. Additionally, the wind stress curl increases significantly as well. These wind changes directly influence the dynamics of Agulhas Leakage and lead to an increase in its transport. Why the resulting

Agulhas Leakage response is lower than in Ivanciu et al. (2022) would need more systematic experiments and analyses, probably also with a larger ensemble number.

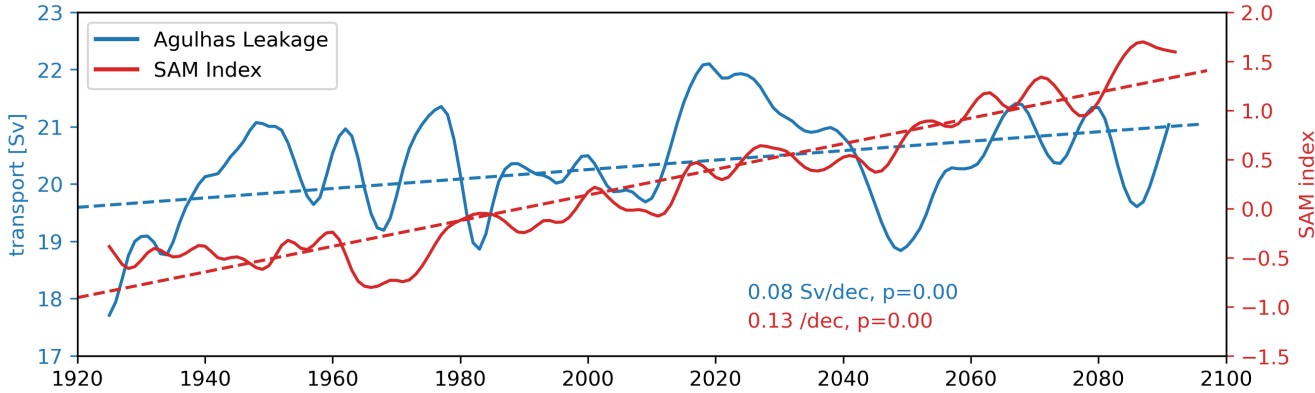

**Figure 8.** Agulhas Leakage and Southern Annular Mode trends for the decadally-filtered ensemble average. Linear trend values and their significance for the period from 1920 to 2100 are also shown.

Looking at the Agulhas Current, we find a strong negative trend under the greenhouse gas forcing, presented in Figure 9. The transport decreases by about $20\,Sv$ by 2100. There is a decrease in the trade winds over the Indian Ocean (Figure A5).



Additionally, the wind stress curl over the southern sub-tropical gyre is decreasing as well. These lead to a decrease in the gyre
strength and its western boundary current, which comes out in the change of the Sverdrup transport at 32°S. This transport
decreases by about $10\,Sv$ during the same time period. While these changes in the wind field can already account for 50% of the
change in the Agulhas Current transport, another source in the Indian Ocean roughly contributes the other half: The transport
through the ITF decreases by up to $8\,Sv$ until the end of the 21st century. These two changes add up to the total decrease in the
Agulhas Current transport. The reduction in Agulhas Current transport ultimately leads to an increase in the Agulhas Leakage
fraction. This means that a higher fraction of the Indian Ocean waters, even though reduced in absolute transport, end up in the
Atlantic and that there is less recirculation.

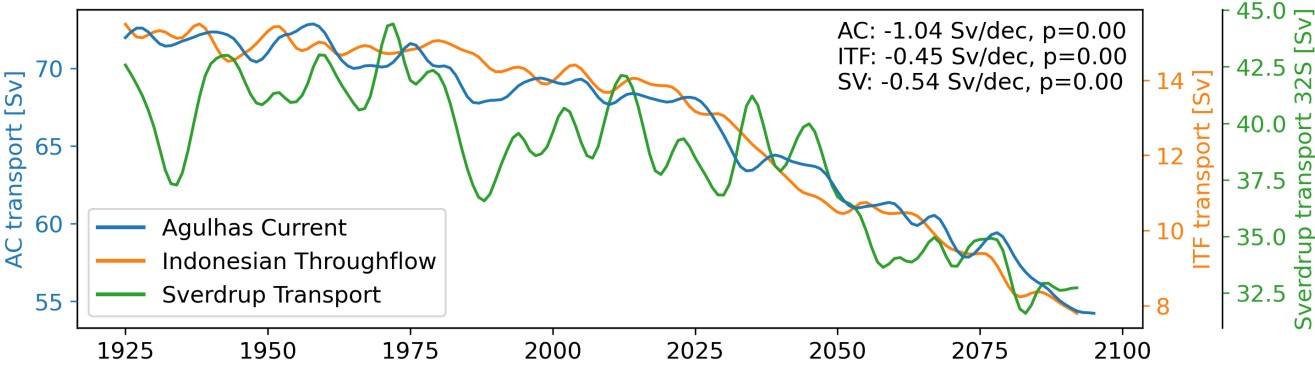

**Figure 9.** Decadally-filtered transport timeseries for the Agulhas Current, the Indonesian Throughflow (ITF), and the Sverdrup transport at
32°S. Dashed lines show the PIcontrol and the full thick lines the ensemble average under transient historical and RCP8.5 $CO_2$ forcing. The
trend values are shown for the period from 1920 to 2100.

We finally investigate what role an increasing Agulhas Leakage might have in the Atlantic Basin, specifically AMOC, in
the future. Figure 10 shows timeseries of the ensemble means of Agulhas Leakage, its salt-flux and the freshwater transport
and AMOC at 34°S. The increase in Agulhas Leakage transport is also associated with a higher salt flux into the Atlantic
Ocean. We find that the salt flux via Agulhas Leakage more than doubles from a pre-industrial mean of $0.24 \pm 0.25\,Sv\,psu$ to
about $0.7\,Sv\,psu$ at the end of the 21st century and the trend specifically starts after 2000. At a similar time, the freshwater
transport at 34°S starts to decrease significantly down to $-0.05\,Sv$ by 2100, as does the AMOC transport down to about $11\,Sv$.
While the AMOC and freshwater decrease is most likely driven by processes in the North Atlantic (Weijer et al., 2019), the
relation of the salt flux to the freshwater transport holds and it enhances the negative trend. This qualitatively describes an
influence of the Agulhas Leakage on AMOC that seems to increase in a warming climate. However, quantifying the impact of
the Agulhas Leakage change ultimately on the AMOC is not straightforward, because so many other factors have an influence
on AMOC. The freshwater transport reaches negative values towards the end of the 21st century, which then implies a positive
salt-advection feedback and an even stronger AMOC decrease in a bi-stable regime.





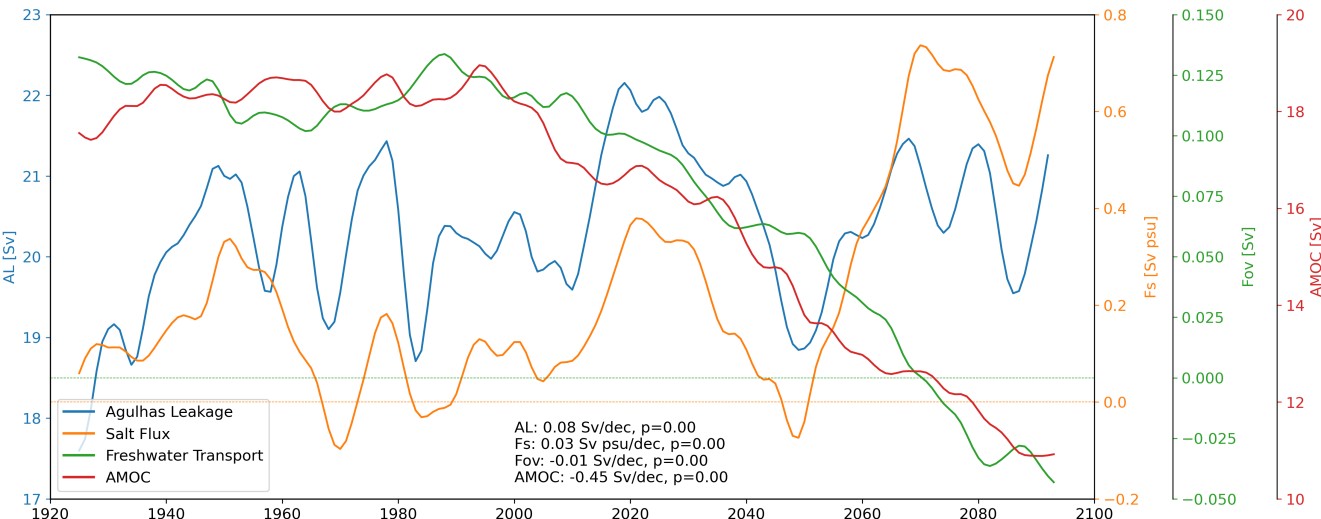

**Figure 10.** Decadally-filtered, ensemble-average timeseries of Agulhas Leakage, Agulhas Leakage salt flux ($F_s$) and AMOC's volume and freshwater transport ($F_{ov}$) at 34°S. Linear trend values and their significance for the period from 1920 to 2100 are also shown.

## 5    Discussion

The interoceanic transport of warm and saline waters by Agulhas Leakage is an important component of the large-scale ocean circulation and therefore global climate. Understanding how Agulhas Leakage varies on longer timescales and how it impacts the Atlantic Ocean in the presence of global warming are thus essential.

Studies so far about the Agulhas Leakage and its variability on long time scales were limited either by the short integration lengths of high-resolution model simulations or by the use of approximations to estimate the Agulhas Leakage transport in mod-

els that do not properly represent the necessary dynamics due to their coarse resolutions. Here, we use a set of high-resolution and fully-coupled Earth system model simulations with the Community Earth System Model, CESM, with a 0.1° ocean (Chang et al., 2020). These simulations are a multi-centennial pre-industrial control run and a three-member ensemble of historical and transient $CO_2$-forcing experiments under an RCP8.5 scenario, the latter for after 2006. A brief evaluation of Sea Surface Height variability against satellite altimetry shows that the model captures the necessary scales in the Agulhas System

as well as globally (Chang et al., 2020). The high resolution allows us to apply a Lagrangian particle tracking algorithm to calculate the Agulhas Leakage transport precisely as the amount of water, that comes from the Agulhas Current and then flows into the Atlantic Ocean through the Good Hope section. Using these model simulations, we investigate the internal variability of Agulhas Leakage on long timescales, its driving mechanisms, and impacts on the AMOC. Additionally, we examine how these relationships might evolve in the future under a warming climate.

Our analysis shows strong interannual variability, which is connected to the formation and propagation of Agulhas rings (Holton et al., 2017). On longer timescales, we find a significant spectral peak at a period of 14 years and strong but not



significant peaks at 40-50 years. These peaks are not robust due to the rather high variability in this region, showing sensitivity to analysis methods and length of the timeseries. It can, of course, be the case that there is no robust internal variability of the system on these timescales. Nevertheless, further research is needed with longer high-resolution simulations.

Our results also show that the Agulhas Leakage is driven by the overlying wind field, in agreement with previous work (Durgadoo et al., 2013), even though the direct mechanism is not quite clear yet (Rühs et al., 2022). Stronger westerlies winds are associated with a larger wind stress curl over the southern Indian Ocean which leads to an increase of the Agulhas Leakage with a time lag of three to four years related to the time scales of baroclinic adjustment in the Indian Ocean. This process is similar to that of DiNezio et al. (2009), who relate transport changes in the Florida Current on interannual to decadal timescales

to a change in the windstress curl to the east of that region. Nevertheless, the region where the winds are important determines the timescale of adjustment processes which could therefore vary between eastern or western parts of a relevant domain. We find the strongest correlation between Agulhas Leakage and the windfield at 50°S and between 40°E and 60°E (Figure A2): A increase in the winds here leads to an increase in Agulhas Leakage. A coherence analysis between the winds and the Agulhas Leakage shows that the winds are the dominant driver of the system across timescales from years to decades.

Our analysis also identifies a significant positive correlation at zero lag for Agulhas Leakage and the Agulhas Current strength. The zero year lag relates to the fact that we define the Agulhas Leakage transport based on the release year of the particles within the Agulhas Current. Therefore, in our simulations, a stronger Agulhas Current leads to a stronger leakage, which is in contrast to the results from (Van Sebille et al., 2009), who found a negative correlation, while (Loveday et al., 2014) described an insensitivity of the Agulhas Leakage to the Agulhas Current's strength. The same analysis but for the AMOC

strength at 34°S reveals a significant positive correlation when Agulhas Leakage leads AMOC by 2 years. This time lag is not consistent with an advective transport timescale because the majority of the particles cross the domain within the first year. Further analysis suggests that the lag is rather controlled by Rossby wave dynamics. We find westward propagating wave signals in the 10°C isotherm depth, which can be used as an indicator of Rossby waves (Olson and Evans, 1986). These waves take about 2-3 years to cross the South Atlantic Ocean, altering the geostrophic balance and consequently the AMOC transport.

This wave connection has been suggested by Biastoch et al. (2008) as well to explain the time scales of interaction between Agulhas Leakage and AMOC. Upon reaching the western boundary, the signal has the potential to be rapidly transported northward through topographic shelf waves and could thereby influence the AMOC further north. However, it is not clear to distinguish between Rossby waves and mesoscale eddies such as Agulhas Rings which propagate at similar speeds. Further analysis are needed here to understand the exact connecting mechanism between Agulhas Leakage and AMOC.

For the global overturning, the heat and salt transports are as important as the transport volume. By tracking the salinity along the particle track, we can evaluate the salt transport of the Agulhas Leakage into the Atlantic. We then investigate the relationship of the salt flux to the freshwater transport at 34°S, which is part of the salt-advection feedback. This feedback describes a connection between the AMOC strength, the freshwater transport at 34°S, and the density difference between the North and the South Atlantic. The freshwater transport is positive in the pre-industrial control run, indicating a stable AMOC.

The real ocean is thought to be in a bi-stable AMOC regime, hence a negative freshwater transport (Rahmstorf, 1996). We note that there are configurations with other modelling frameworks that have a bi-stable AMOC, and Deshayes et al. (2013)





show that stability also relates to model resolution with higher resolution leading to a negative freshwater transport. However, to complicate matters further, such AMOC stability varies between forced ocean-only experiments and fully-coupled Earth system models as well (Cheng et al., 2018). Van Westen and Dijkstra (2023) show that many coupled models have errors in
their surface salinities due to atmospheric precipitation biases which then influences the freshwater transport in the ocean. They also show that models from the Climate Model Intercomparison Project phase 6 (CMIP6) have both positive and negative freshwater transports, but the ones with a negative transport underestimate the AMOC transport; no model gets both in the observational range. In this study we were able to identify a strong negative correlation between the salt flux and the freshwater transport across 34°S, suggesting that in this CESM configuration, the salt input from the Agulhas Leakage therefore influences
the stability of the AMOC through the salt-advection feedback.

The 3-member ensemble of transient simulations under historical and RCP8.5 forcing shows an increase in the Agulhas Leakage transport in response to the anthropogenic global warming, inline with previous study that also found stronger Agulhas Leakage transport under a warmer climate. The increase we find is not as strong as some other studies show or suggest (Ivanciu et al., 2022; Beech et al., 2022), but still significant. The reason for the modest increase in Agulhas Leakage transport
in our simulations lies in competing effects from the global circulation. On one hand, the wind field favours an enhanced Agulhas Leakage transport through strengthened and southward shifted westerlies, as indicated by the strengthening in Southern Annular Mode. On the other hand, the projected weakening of the Agulhas Current transport, is conductive of of weakening of the Agulhas Leakage transport, by reason of the positive correlation between Agulhas Leakage and the Agulhas Current previously identified in our analysis. The Agulhas Current is further impacted by Indian Ocean sources, both originating within and
also brought in from the Pacific Ocean by the ITF (Durgadoo et al., 2017). Our results show that two factors almost equally contribute to the decline in the Agulhas Current: A reduction of the trade winds in the Indian Ocean contributing to a weaker western boundary current and a weaker ITF associated with reduced inflow of relatively warm and fresh water from Pacific Ocean origin into the Indian Ocean. The reduction of the ITF, and in consequence of the Agulhas Current transport, can be seen as a weakening of the global overturning, probably related to thermohaline causes. In another study, Hu et al. (2021)
performed hosing experiments to force an AMOC collapse, which then leads to a reduction of $15\,Sv$ in the Agulhas Current and $10\,Sv$ in the Indonesian Throughflow in the first 100 years after the freshwater release, therefore similar magnitudes as we found. This further underlines that the whole Agulhas system is controlled by the global overturning circulation as well as local wind-driven dynamics.

Regarding the future evolution of the AMOC, we also see an increase in the salt transport of the Agulhas Leakage. We
calculate the salt flux as the difference of the particle salinities to the mean salinity at the crossing section. Van Westen and Dijkstra (2023) investigated the similar model runs, but at lower horizontal resolution, and found a change in the local salinity at the section during global warming due to more evaporation than precipitation, which then impacts the reference salinity for the salt flux estimation. However, when fitting a linear trend to the section and using this as a reference instead of just a mean, we do find a significant increase of the salt flux as well but with a smaller magnitude. As the salt flux trend is the strongest
after 2000, opposite to the Agulhas Leakage trend, it seems that this is based on a change in the hydrography rather than the volume transport. The increased salt flux then contributes to a decrease of the freshwater transport, which ultimately could





decrease the AMOC stability within the salt-advection-feedback theory. Van Westen and Dijkstra (2023) also show that the decrease in the freshwater flux is salinity based and mostly depends on changes in the upper 1500m, which is consistent with the impact of the Agulhas Leakage salt flux. The decrease evolves even to the point of a negative freshwater transport at the end of the 21st century and could lead to a possible AMOC collapse in the future. On the other hand, hypothesis exist that the salt input from the Agulhas Leakage into the Atlantic ultimately reaches the North Atlantic and deep water formation regions (Weijer and van Sebille, 2014). The salt can then play a role in setting the local stratification and thereby positively impacting deep water formation. However, quantifying these processes and the ultimate impact on the AMOC strength needs further research. For this purpose, one would need ensemble experiments with eddy-rich Atlantic-wide or global configurations and long experiments with predicted or idealised freshwater hosing.

## 6    Conclusions

A set of unprecedentedly long high-resolution CESM simulations allowed us to perform an extended analysis of the Agulhas Current system and the interoceanic transports associated with Agulhas leakage. In the pre-industrial climate, our analysis shows that Agulhas Leakage variability is driven by variation in the wind field over the southern Indian Ocean, in agreement with earlier findings. Furthermore, we show that Agulhas Leakage is related to the strength of the Agulhas Current, with a stronger current leading to enhanced Agulhas Leakage transport. Agulhas Leakage is found to impact the AMOC through Rossby wave dynamics and/or mesoscale eddies that propagate into the Atlantic Ocean. The salt flux of Agulhas Leakage into the Atlantic influences the meridional freshwater transport by the AMOC and consequently AMOC stability through the salt-advection feedback. Agulhas Leakage is projected to increase under global warming due to strengthening of the westerly winds in the Southern Hemisphere in the CESM 3-member ensemble of high-resolution transient simulations. However, the projected Agulhas Leakage increase in our simulations is weaker than previously reported. This weaker increase in Agulhas Leakage is imputable to the combination of a strong reduction of the Agulhas Current transport which is the result of both a decreased wind-related Sverdrup transport, and a reduction of the Indonesian Throughflow. Nevertheless, the moderate Agulhas Leakage increase under a warming climate is accompanied by an increasing salt flux that then reduces the stability of the AMOC. The reduced stability combined with the weakening of the strength of AMOC under a warming climate exacerbate the potential of a future AMOC collapse. This study sketches a new picture of the role of Agulhas Leakage in the climate system through a fully coupled eddy-rich earth system model. It also highlights points that will need further research attention, such as the exact connecting mechanism between Agulhas Leakage and AMOC and the relevance of the increased salt input into the Atlantic Ocean on AMOC stability in comparison to other factors in play.

*Code and data availability.* The data and material that support the findings of this study are available through GEOMAR at https://hdl.handle.net/20.500.12085/f10e76e5-0e1e-4dee-95b5-45d6275eb144 (Grosselindemann et al., 2024).



*Author contributions.* AB and DB initiated and designed the analysis. HG led the paper and wrote and structured the manuscript. HG performed all the analyses under supervision from all authors. All authors discussed the analyses and provided comments to the text.

*Competing interests.* The authors declare no competing interests.

*Acknowledgements.* The high-resolution CESM simulations were initiated by the International Laboratory for High-Resolution Earth System Prediction – a collaboration between the Qingdao Pilot National Laboratory for Marine Science and Technology, Texas A&M University, and the US National Science Foundation (NSF) National Center for Atmospheric Research (NCAR). A major portion of the high-resolution CESM simulations were completed on Frontera at the Texas Advanced Computing Center (TACC) of the University of Texas at Austin, TX, US, under project number ATM20005. F.C. was partly supported by the NSF grant AGS-2231237. This material is based upon work
supported by the NSF NCAR, which is a major facility sponsored by the NSF under Cooperative Agreement 1852977. HG was supported by a scholarship of the Prof. Dr. Werner Petersen Stiftung and a DAAD PROMOS fellowship from the Christian-Albrechts University in Kiel.

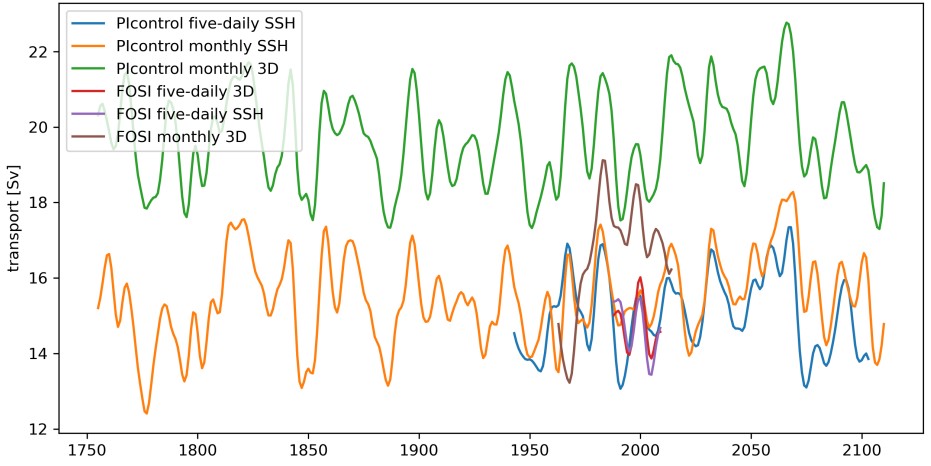

**Figure A1.** Decadally-filtered transport time series of Agulhas Leakage from all approaches. See Section 3.1 for details.





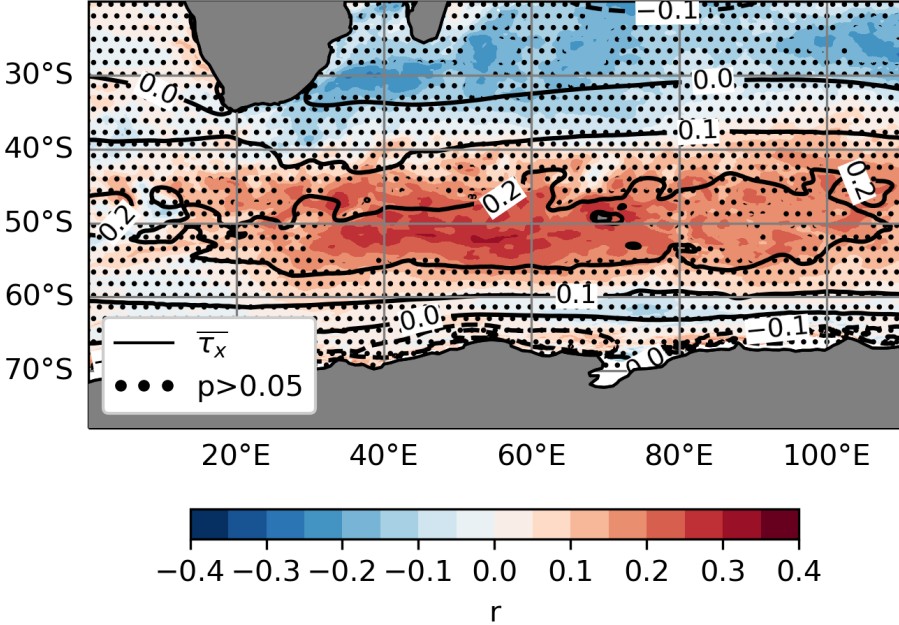

**Figure A2.** Correlation of Agulhas Leakage transport with the annual-mean zonal windstress at each grid point for PIcontrol. Correlations are not significant in the dotted regions. Contour lines show the mean zonal windstress in N m$^{-2}$ and the big black rectangle outlines the area used to obtain the maximum zonal windstress for the time series analysis.



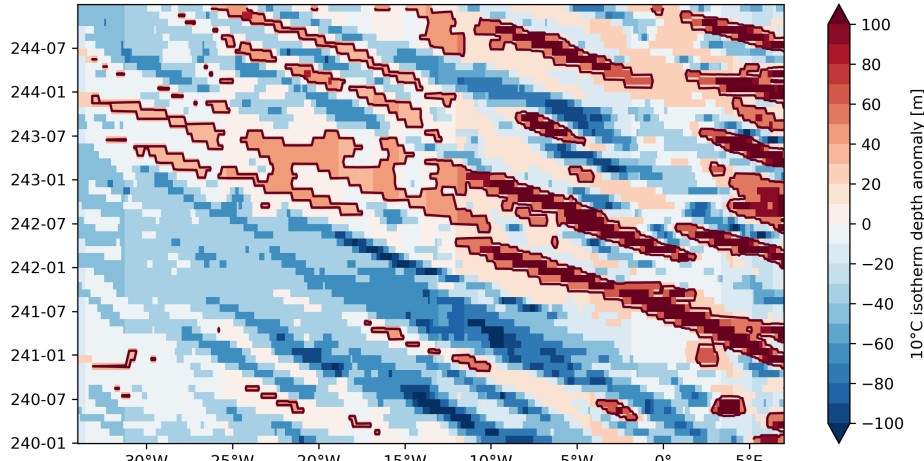

**Figure A3.** Timeseries of the 10°C isotherm depth anomaly from PIcontrol for years 240 to 245 as a function of longitude and at 34°S. The 30-m depth is shown as a contour line.





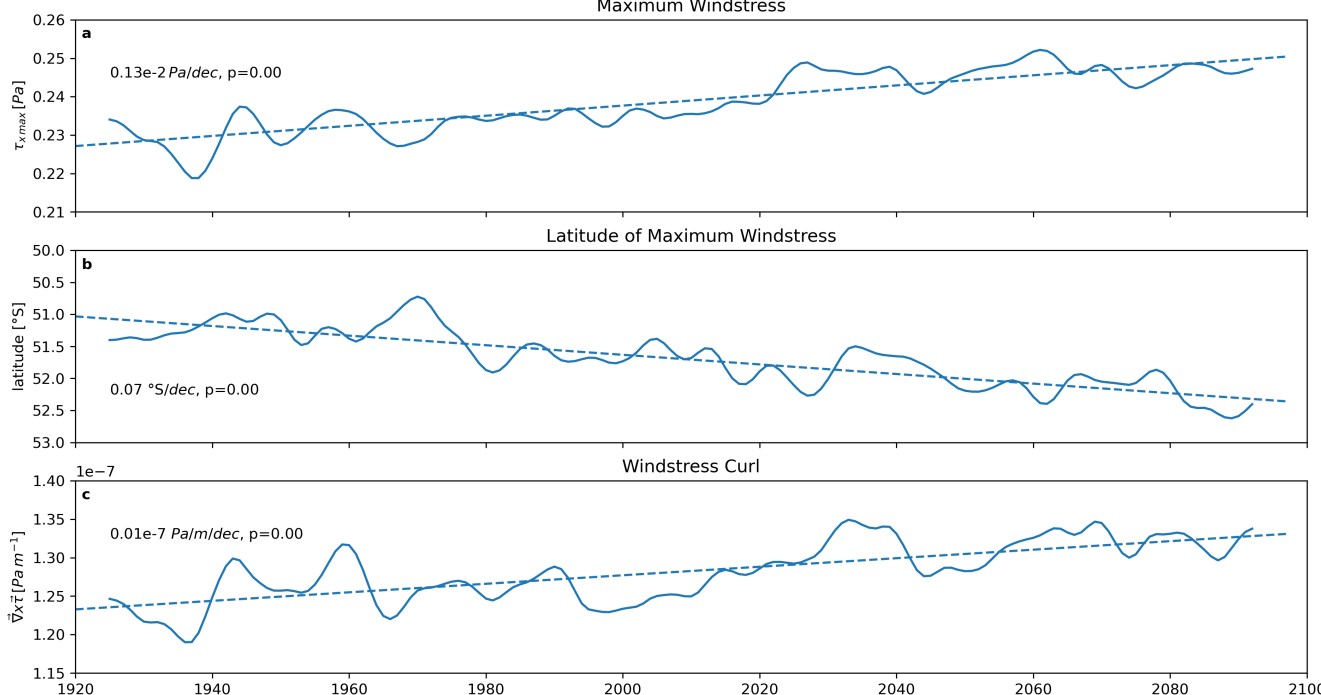

**Figure A4.** Timeseries and trends for the period from 1920 to 2100 maximum zonal wind stress $\tau_{x\,max}$ (a), latitude of the maximum zonal windstress in °S (b), and average wind stress curl $\nabla \times \tau$ (c) as decadally-filtered ensemble means, trends are calculated on unfiltered data.



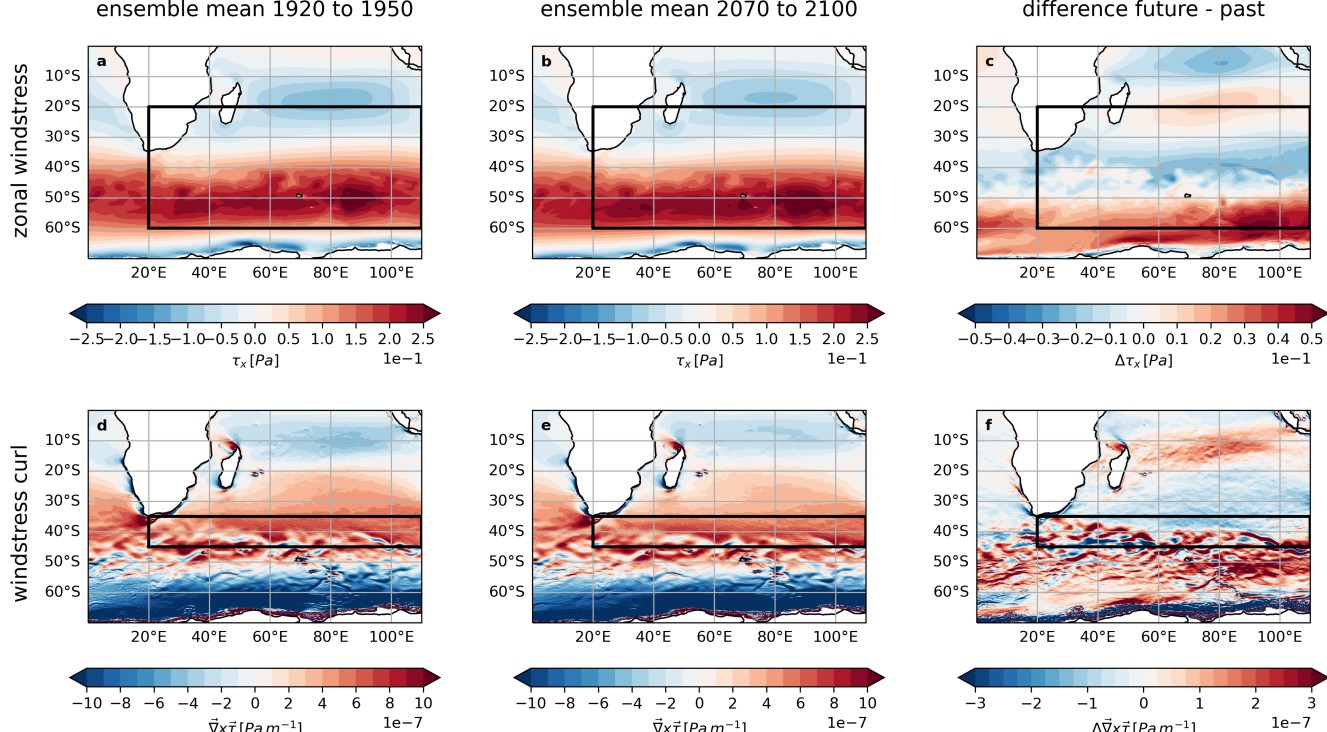

**Figure A5.** Ensemble-averaged zonal windstress $\tau_x$ (a-c) and windstress curl $\nabla \times \tau$ (d-f) historical (a,d) and future (b,e) mean states for the periods from 1920 to 1950 and 2070 to 2100, respectively, and their change (c,f), the periods were chosen to represent a mean climate state at the begin and end of the simulation period, boxes show areas used to calculate wind metrics for Figure 5 and Figure A4



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
