# Peer review of "Long-term Variability and Trends of Agulhas Leakage and its Impacts on the Global Overturning"

_EGUsphere, 2024_

## Referee Comment (RC2)

In this study the authors analyse the role of Agulhas Leakage (AL) on the Atlantic Meridional Overturning Circulation (AMOC). They analyse the strongly eddying version of the Community Earth System Model (CESM, CMIP5 version) from the iHESP project. This high-resolution version of the CESM is needed to realistically capture Agulhas Current dynamics. They analyse two experiment: the pre-industrial control simulation and the historical forcing (1850 − 2005) followed by RCP8.5 (2006 − 2100).

The authors first analyse the drivers, variability and trends in Agulhas Leakage. There is a direct wind effect on AL changes, but other far-field contributions are also important such as Indonesian Throughflow on the Agulhas Current strength. The next step is to link AL changes to the AMOC, this is done by analysing the AL-induced freshwater (or salinity) transport along 34°S in the Atlantic Ocean. The freshwater transport carried by the AMOC, indicated by $F_{\mathrm{ovS}}$, is an important indicator for AMOC stability. When AMOC carries net salinity into the Atlantic basin ($F_{\mathrm{ovS}} < 0$), the salt-advection feedback amplifies freshwater perturbations and destabilise the AMOC. The authors show in their last step is that the AL contributes to a greater salinity transport into the Atlantic Ocean under climate change, hence the AL influences the AMOC stability.

I would like to thank the authors for their interesting study. The manuscript is well written, clearly visualised, the analyses are well conducted and (mostly) correctly interpreted. I have a few (major) remarks on the AMOC stability indicator (the $F_{\mathrm{ovS}}$) and the link with an increased AL salinity transport. The comments below need to be addressed before I recommend the manuscript for publication.

Major comments and suggestions:

1. The parts of the manuscript which discuss the $F_{\text{ovS}}$ changes from AL and links with AMOC stability need to be more carefully stated. The interpretation is not always correct and a few arguments are missing, see the following points:

    - The AMOC carries relatively salty water northward in the North Atlantic Ocean, the local $F_{\text{ov}}$ is negative (e.g., at 40°N, see Jüling et al., 2021). When a freshwater perturbation is applied in the North Atlantic Ocean (in a hosing set-up), the AMOC strength and associated salinity transport reduce. The reduced salinity transport may amplify the original freshwater perturbation, leading to an even greater freshwater perturbation and further decreasing the AMOC strength: the (positive) salt-advection feedback. This feedback is only effective (see section 4b in Huisman et al., 2010) when velocity-induced and salinity-induced freshwater transport changes (under a freshwater perturbation) do not oppose each other. This is only the case when the AMOC carries net salinity into (exports net fresh water out of) the Atlantic basin, and hence $F_{\text{ovS}} < 0$. For the case when $F_{\text{ovS}} > 0$, the North Atlantic freshwater perturbations are usually 'flushed out' of the Atlantic Ocean and there is no positive salt-advection feedback. So the quantity $F_{\text{ovS}}$ only represents whether the AMOC amplifies (North Atlantic) freshwater perturbations, this is mentioned by the authors (line 353).

      The study by Haines et al. (2022) questions whether the $F_{\text{ovS}}$ is a useful metric for AMOC stability analysis in fully-coupled climate models (under constant pre-industrial conditions). They show that $F_{\text{ovS}}$ changes hardly influence the North Atlantic freshwater transport and a North Atlantic freshwater change is needed to modify the AMOC strength (Rahmstorf, 1996). However, van Westen et al. (2024a) demonstrated that the $F_{\text{ovS}}$ is a useful metric for AMOC stability analysis in the (low-resolution) CESM and this was consistent with previous work (e.g., Huisman et al., 2010). The differences between Haines et al. (2022) and van Westen et al. (2024a) could be related to the magnitude of the freshwater perturbations, where the latter study varies a North Atlantic freshwater flux forcing between 0 and 0.66 Sv.

Relatively small freshwater/salinity perturbations at 34°S may be ineffective in modifying the North Atlantic Ocean freshwater content. This doesn't imply that there are no relations between AMOC strength and $F_{\mathrm{ovS}}$ (e.g., Figure 8a in van Westen and Dijkstra, 2024). The $F_{\mathrm{ovS}}$ is also positive in the CESM (before 2070, Figure 10) and in this regime it is not very likely that AL changes destabilise the AMOC. The authors could argue that a greater AL salinity transport under climate change is preconditioning the AMOC to a more sensitive regime. You could also use the arguments that the CESM has known freshwater transport biases (van Westen and Dijkstra, 2024) and the observed $F_{\mathrm{ovS}}$ is negative (Arumí-Planas et al., 2024). My main point here is that the conclusions drawn from the AL changes on AMOC stability (e.g., lines 468 – 469) are sometimes strongly phrased. These parts need to be revised and a better discussion on the role of $F_{\mathrm{ovS}}$ is needed (in both the introduction and discussion).

- To continue with my previous point: the interpretation of $F_{\mathrm{ovS}}$ changes under strong transient responses. The quantity $F_{\mathrm{ovS}}$ can only be used under (quasi-)equilibrium conditions (Rahmstorf, 1996) and these conditions include the equilibration of the Atlantic Ocean interior. Under strong climate change (RCP8.5) or large freshwater flux changes (Oríhuela-Pinto et al., 2022; Jackson et al., 2022) the $F_{\mathrm{ovS}}$ responses and its effect on AMOC stability are much more difficult to interpret. It should be noted that the AMOC responses under climate change are dependent on the initial/historical $F_{\mathrm{ovS}}$ value (Liu et al., 2017; van Westen and Dijkstra, 2024). The AL influences the $F_{\mathrm{ovS}}$ under climate change (Figure 10) and directly connecting this to AMOC stability is difficult, as we consider the transient case. Once the CESM is equilibrated to the new 2100 radiative forcing conditions, the more negative $F_{\mathrm{ovS}}$ suggests that the AMOC is closer to its tipping point, but this can't be verified from the transient results. Again, a more careful interpretation is needed when analysing the transient $F_{\mathrm{ovS}}$ and AMOC responses.

- Lines 66 and 357: The authors suggest that a negative $F_{\mathrm{ovS}}$ (in the AMOC on state) allows for a bi-stable AMOC regime. This is indeed the case in many conceptual (AMOC) models and models of intermediate complexity (Dijkstra 2024). However, the recent quasi-equilibrium hysteresis hosing simulation performed with a low-resolution version of the CESM (van Westen and Dijkstra, 2023) indicates that positive $F_{\mathrm{ovS}}$ values (in the AMOC on state) are also part of the bi-stable AMOC regime. Sea-ice feedbacks, which were poorly captured in idealised models, modify the AMOC hysteresis behaviour (van Westen et al., 2024b) and allow for positive $F_{\mathrm{ovS}}$ values to be part of the bi-stable AMOC regime. Climate model biases also shift the saddle-node bifurcations and negative $F_{\mathrm{ovS}}$ does not exclusively indicate the bi-stable AMOC regime (Dijkstra and van Westen, 2024). Recent work by Lohmann et al. (2024) demonstrated a multi-stable AMOC regime under varying freshwater flux forcing, so not only bi-stable.

2. Line 227: The total freshwater transport at 34°S can be decomposed (Jüling et al., 2021) into four different contributions: overturning, azonal (gyre), barotropic ($\approx 0$) and eddies. I would argue that Agulhas rings, which are part of the AL, would end up in the eddy component and not (directly) in the overturning component. Have you considered determining the eddy-induced freshwater transport by AL? In Jüling et al. (2021) there is a negative freshwater transport trend in the eddy component (their Figure 7). I agree with the authors that oceanic adjustment by Agulhas rings or Rossby waves can eventually influence the overturning component (line 297), but this is relevant on time scales longer than one year.

3. A low-resolution (1°) companion CESM simulation is available within the iHESP project (line 120). Climate model projections at 34°S (and elsewhere) are model resolution dependent (van Westen and Dijkstra, 2024) and could be relevant in the AL responses. Such a resolution comparison is also useful when considering other (1° resolution) CMIP6 models. I was wondering why the low-resolution CESM was not included in the analysis, at least a clear motivation is missing.

   I would like to encourage the authors to conduct the analysis on the most interesting quantities in the low-resolution version of the CESM

from the iHESP project. For example, it would be very interesting to see Figures 9 and 10 for the low-resolution CESM and to compare against the high-resolution CESM results. The low-resolution CESM results should go in the Appendix and are discussed in the main text. I'm not asking for a complete low-resolution CESM nor CMIP6 analysis, this is too much work and beyond the scope of the paper. The high-resolution CESM results are (and should be) central here.

Minor comments and suggestions:

1. Lines 1, 20, 25, and throughout manuscript: 'the warm and salty waters'. Refer to the *relatively* warm and salty waters (or quantify warm and salty waters). Please check and fix throughout manuscript.

2. Line 41: Maybe helpful for the reader to provide the time-mean formation rates and propagation speeds of the Agulhas 'eddies'.

3. Throughout manuscript: Perhaps use Agulhas *rings* instead of Agulhas *eddies*. Ocean eddies arise from baroclinic instabilities, while ring shedding is a different processes.

4. Line 58: The salt-advection feedback is the dominant destabilising AMOC mechanism and has been demonstrated in climate/AMOC models of varying complexity (Dijkstra, 2024). This sentence suggests that there are more (equally important?) feedback mechanisms, which ones did you consider? Note that the AMOC can be weakened through rapid climate change (e.g., Gérard and Crucifix, 2024) or large freshwater flux change (Oríhuela-Pinto et al., 2022), but these AMOC responses are related to the imposed forcing and not to a self-amplifying feedback loop (see also Major point 1).

5. Line 67: '... lead to an AMOC collapse (Rahmstorf, 1996)'. I would add the study by van Westen et al., (2024a), they demonstrate an AMOC collapse in a modern complex climate model under quasi-equilibrium hosing conditions. The study by Dijkstra (2024) is also relevant here, as it provides a review of AMOC tipping behaviour in a hierarchy of climate models.

6. Line 97 – 98: Only one RCP scenario is available within the iHESP project, namely the RCP8.5 scenario. This suggests that more RCP

scenarios are available for the CESM and you only selected the RCP8.5 for the analysis. Please refer to the RCP8.5 scenario here (or motivate why you have focussed on the RCP8.5 scenario).

7. Line 120: 'lower resolution counterpart' → lower (1°) resolution counterpart. Good to quantify the resolution here.

8. Lines 127, 151, 161, etc.: 'absolute dynamic topography (Sea Surface Height above geoid)'. I would recommend to use 'dynamic sea level (DSL)', following the terminology proposed by Gregory et al. (2020). The DSL corresponds to the 'SSH' variable from the CESM output.

9. Line 128: 'model's capacity', which model are you referring to? The FOSI?

10. Line 129: This product → The altimetry product. The reference to 'this product' is not clear to me.

11. Line 207: Figure A5, maybe re-order the appendix figures so that they appear in their reference order as in the main text. So Figure A5 → Figure A2 (and move/re-label these figures in the Appendix).

12. Line 226: salt flux and freshwater flux. I suggest to use salt *transport* and freshwater *transport*, which is then consistent with the definition used in line 227 (annual freshwater transport). Check this throughout the manuscript.

13. Line 244: Upsream → Upstream (typo)

14. Line 247: In the model → In the CESM (mention the CESM here).

15. Line 264: Is this the variability between 2 – 5 years, the period with significant (95%-Cl) peaks? Good to quantify the 'inter-annual variability' here.

16. Line 265: Alternatively, you could determine the confidence level at which the 40–50 year period is significant (similar to the p-value of 0.06, line 169).

17. Line 405: transport volume → volume transport

18. Line 410: A recent study by Arumí-Planas et al. (2024) quantified the $F_{\mathrm{ovS}}$ from available observations, the present-day $F_{\mathrm{ovS}}$ is indeed negative. This study is worth mentioning here.

References:

1. Arumí-Planas et al., (2024): A Multi-Data Set Analysis of the Freshwater Transport by the Atlantic Meridional Overturning Circulation at Nominally 34.5S, https://doi.org/10.1029/2023JC020558

2. Dijkstra (2024): The role of conceptual models in climate research, https://www.sciencedirect.com/science/article/pii/S016727892300338X

3. Dijkstra and van Westen (2024): The Effect of Indian Ocean Surface Freshwater Flux Biases On the Multi-Stable Regime of the AMOC, https://a.tellusjournals.se/articles/10.16993/tellusa.3246

4. Jackson et al., (2022): Understanding AMOC stability: the North Atlantic Hosing Model Intercomparison Project, https://doi.org/10.5194/gmd-16-1975-2023

5. Gérard and Crucifix (2024): Diagnosing the causes of AMOC slowdown in a coupled model: a cautionary tale, https://doi.org/10.5194/esd-15-293-2024

6. Gregory et al., (2020): Concepts and Terminology for Sea Level: Mean, Variability and Change, Both Local and Global, https://link.springer.com/article/10.1007/s10712-019-09525-z

7. Haines et al., (2022): Variability and Feedbacks in the Atlantic Freshwater Budget of CMIP5 Models With Reference to Atlantic Meridional Overturning Circulation Stability, https://doi.org/10.3389/fmars.2022.830821

8. Huisman et al., (2010): An Indicator of the Multiple Equilibria Regime of the Atlantic Meridional Overturning Circulation, https://journals.ametsoc.org/view/journals/phoc/40/3/2009jpo4215.1.xml

9. Jüling et al., (2021): The Atlantic's freshwater budget under climate change in the Community Earth System Model with strongly eddying oceans, https://doi.org/10.5194/os-17-729-2021

10. Liu et al., (2017): Overlooked possibility of a collapsed Atlantic meridional overturning circulation in warming climate, https://www.science.org/doi/10.1126/sciadv.1601666

11. Lohmann et al., (2024): Multistability and intermediate tipping of the Atlantic Ocean circulation, https://www.science.org/doi/full/10.1126/sciadv.adi4253

12. Oríhuela-Pinto et al., (2022): Interbasin and interhemispheric impacts of a collapsed atlantic overturning circulation, https://doi.org/10.1038/s41558-022-01380-y

13. Rahmstorf (1996): On the freshwater forcing and transport of the Atlantic thermohaline circulation, https://link.springer.com/article/10.1007/s003820050144

14. van Westen and Dijkstra (2023): Asymmetry of AMOC Hysteresis in a State-Of-The-Art Global Climate Model, https://doi.org/10.1029/2023GL106088

15. van Westen and Dijkstra (2024): Persistent climate model biases in the Atlantic Ocean's freshwater transport, https://doi.org/10.5194/os-20-549-2024

16. van Westen et al., (2024a): Physics-based early warning signal shows that AMOC is on tipping course, https://www.science.org/doi/full/10.1126/sciadv.adk1189

17. van Westen et al., (2024b): The Role of Sea-ice Processes on the Probability of AMOC Transitions, https://arxiv.org/abs/2401.12615

---

## Author Comment (AC1)

Review of: Long-term Variability and Trends of Agulhas Leakage and its Impacts on the Global Overturning", by Großelindemann et al.

In this paper, the authors analyze Agulhas Leakage and its drivers and impacts in a suite of eddy-resolving coupled climate simulations. They find that Agulhas Leakage is well represented by the model, according to several metrics. They find positive correlations between Agulhas Leakage and several metrics of wind stress, and find that this model projects an increase in Agulhas Leakage in response to an aggressive forcing scenario, despite a weakening of the Agulhas Current itself.

I found the paper very well written, and a pleasure to read. The methodology is clearly described, the analysis is convincing, and the results are interesting and relevant. I do have a list of minor comments, which will require only minor revisions.

> We are delighted that the reviewer received our manuscript very positively. We would like to thank him for his time reviewing the manuscript and providing very constructive comments and suggestions.

l. 125: employes -> employs

> Changed.

Fig. A1: I think it would be better to make subplots for the piControl and FOSI simulations. It is hard to see the individual curves representing the FOSI simulations. Besides, the variability in the two sets should not be expected to be identical, so combining them in the same plot does not make much sense. This plot is important, as it allows the reader to judge the accuracy of the method applied. Despite the muddled mess, the three FOSI curves don't seem to track each other very well.

> We agree with the reviewer and have changed the figure to have subplots. There is a very slight temporal shift between the FOSI monthly and the 5-daily time series, which is reflected in the correlation values we use to judge the methods. The shift in absolute transport values is discussed in the text as well.

Fig A2: This figure does not show a black rectangle, as the caption claims.

> This has been corrected, while the appendix figures were reordered.

L. 158: plausible -> would accurate be a better word here?

> We think that "plausible" better reflects what we want to convey. 'accurate' may imply that we could somehow precisely measure the difference between the exact transport in the model and our estimate.

ll. 207-208, 233-235: Duplicative.

Removed it from l.212ff, as that paragraph is more about winds and the second about circulation.

l. 227 and elsewhere: Fov is the freshwater flux /induced by the overturning circulation/.

Good point, we added this information wherever it is first introduced in each section.

l. 244: Uptream -> Upstream

Changed.

l. 265, 285, 276: Unless you can find a way to make a more robust significance estimate of this spectral peak at 14 years, I would not put that much emphasis on it; especially if it is only one estimate that sticks out, instead of a few adjacent estimates. At 95% confidence level, one is to expect 5 'false positives' for every 100 measurements.

Given the high level of intrinsic variability, we think that it is worth mentioning the spectral peak at 14 years, even though it is only significant at 95% confidence.

l. 273: So minimum zonal wind stress represents the easterlies in the subtropical belt?

Yes, we have added the following sentence to the methods section: "The maximum and minimum zonal wind stress represent the Southern Hemisphere Westerlies and the Subtropical Easterlies, respectively.", l. 211

l. 283: Filtering might also play a role in spreading out the signal.

Good point. We have added the following sentence: "Barotropic and baroclinic adjustment processes as well as filtering of the time series can explain the 0-3-year spread in the range of lead times (Anderson et. al 1977)." l. 289

Figure 6: I find the correlations between Fov and AL suspicious, as they are significant stronger than -0.4 for lags between +/- 7 years. What does that mean? Do both time series have decorrelation times of several decades? The paper says that the time series have been detrended, can you confirm?

The Fov time series shows an increasing trend in the first ~100 years, reaches a maximum, and then decreases until the end due to model drift and adjustment times. We have detrended it, but a linear trend does not completely get rid of the non-linear long-term variation. However, the peak at three years is robust when we calculate the correlation for the increasing and decreasing parts separately. But to be consistent with the other metrics, we chose to show the values from the full time series calculation. We added a short paragraph about this in that part of the manuscript as follows:

"We note that the Fov timeseries in the model has an increasing trend in the first 100 years and then decreases again. This impacts the detrending of the timeseries before calculating the correlations and leads to high correlations across all lead times. The peak

at a 3-year lag remains robust even when increasing and decreasing parts of the time series are considered separately. " l.320

l. 294 and beyond: I'm wondering if this argument could be taken one step further by actually calculating an appropriate east-west gradient (upper ocean pressure, SSH, or maybe even the depth of the 10 degree isotherm) and comparing that to AMOC strength at 34S.

We have tried the reviewer's suggestion, but did not find any meaningful connections between a simple SSH-gradient and AMOC strength at this latitude. Properly calculating a gradient is not straightforward, and since there is no meaningful signal from a simple calculation and an AL - Rossby wave/eddy connection is also vague, we have concluded that a more detailed analysis would be beyond the scope of the manuscript.

309: stable -> stabilizing, negative?

Yes, we have modified the text as follows: "We find a mean northward freshwater transport of 0.1 ± 0.03 Sv over the simulation years 150-520 which remains positive during this entire time, indicating a stabilising salt-advection feedback." l. 314

l. 321: I think it is important to mention the trend over that same period for the control simulation, since I suspect it is not much smaller than that of the historical + future simulations. If even the control has a significant trend over that period, then I suspect that that will modify the conclusion.

The control has an insignificant trend of 0.01 Sv per decade with a p-value of 0.42. We have added this information to the text and the related figure as follows:"The PIcontrol shows a small, statistically insignificant trend of only 0.01 Sv/dec." l. 336

l. 362: are -> is. Essential for what?

Rephrased it as follows: "Investigating how Agulhas leakage varies on longer timescales is thus essential to understand its impacts on the Atlantic Ocean in the presence of global warming." l.374

l. 358: It may be worth noting that the Fov as a metric of the salt-advection feedback would asymptote to zero (from a positive value) upon a decreasing AMOC. The fact that it crosses zero suggests changes in the stratification at 34S, giving credence to the conclusion here. Is there a way to quantitatively compare Fov with the salt flux induced by AL?

The suggested analysis would need a closed salt budget calculation. The domain we track the particles in is not big enough to track all the particles that cross 34S at some point during their lifetime. We therefore do not have a complete inventory of all the particle salt transports directly at 34S and hence cannot really calculate the salt budget.

Additionally, the other reviewer correctly mentioned that "The quantity FovS can only be used under (quasi-)equilibrium conditions (Rahmstorf, 1996) and these conditions include the equilibration of the Atlantic Ocean interior. Under strong climate change (RCP8.5) or large freshwater flux changes (Orihuela-Pinto et al., 2022; Jackson et al., 2022) the FovS responses and its effect on AMOC stability are much more difficult to interpret." We thus refrain from over-interpreting the Fov-AL relation in our transient experiments. Instead we have more carefully formulated our conclusions in this respect. Please view our comments to reviewer Van Westen.

l. 393: Correct parentheses around reference.

Done.

l. 397: This conclusion is more or less unless contradicted a few lines later, where it is claimed that no distinction can be made between Rossby waves or propagating rings. I don't see it as a problem that we don't quite know the dynamical character of these propagating signals (maybe they are one and the same!), but it would be good to be consistent. Also, l. 303 acknowledges a potential role for winds, which is missing here.

In response to the reviewer's comment, we have changed the phrasing from "is controlled by Rossby wave dynamics" to "could be controlled… ". With this change, we think it is fine to discuss that statement in the following sentences. We have also added mesoscale eddies as an extra possibility and not just as a description for Agulhas Rings in lines 309 and 417 . We agree with the comment on the role of winds. We moved the wind-related discussion from the results to the discussion section: It fits better there.

l. 403: It may be semantics, but I'm not sure if Agulhas Rings can be classified as a mesoscale eddies. In my mind, they have a different character and dynamical origin, not in the least because of the barotropic nature of rings that contrasts with the baroclinic character of eddies.

Please see the above response.

l. 427: conductive -> conducive. Correct double 'of'.

Corrected.

ll. 468-469, 16-17, 450, and other places: I'm uncomfortable with the strong statements that are made regarding the links between Agulhas Leakage and the stability of the AMOC and potential for collapse. Even though I think that the link between Fov and AMOC (bi-)stability is a compelling theory, there is still a lot of work to do to confirm this theory (for instance, by demonstrating it in an eddy-resolving climate model). I personally would not go beyond a statement along the lines of 'with potential implications for the stability of the AMOC'.

You're right, we should be more careful: We have adjusted our phrasing accordingly in various places in the manuscript. The other reviewer had similar

comments about this as well. We now just say that our analysis suggests that there could be a connection or a potential role of the salt transport of the Agulhas leakage to AMOC stability, but that further research is needed here.

---

## Author Comment (AC2)

In this study the authors analyse the role of Agulhas Leakage (AL) on the Atlantic Meridional Overturning Circulation (AMOC). They analyse the strongly eddying version of the Community Earth System Model (CESM, CMIP5 version) from the iHESP project. This high-resolution version of the CESM is needed to realistically capture Agulhas Current dynamics. They analyse two experiment: the pre-industrial control simulation and the historical forcing (1850 – 2005) followed by RCP8.5 (2006 – 2100).

The authors first analyse the drivers, variability and trends in Agulhas Leakage. There is a direct wind effect on AL changes, but other far-field contributions are also important such as Indonesian Throughflow on the Agulhas Current strength. The next step is to link AL changes to the AMOC, this is done by analysing the AL-induced freshwater (or salinity) transport along 34°S in the Atlantic Ocean. The freshwater transport carried by the AMOC, indicated by $F_{ovS}$, is an important indicator for AMOC stability. When AMOC carries net salinity into the Atlantic basin ($F_{ovS}$ < 0), the salt-advection feedback amplifies freshwater perturbations and destabilise the AMOC. The authors show in their last step is that the AL contributes to a greater salinity transport into the Atlantic Ocean under climate change, hence the AL influences the AMOC stability.

I would like to thank the authors for their interesting study. The manuscript is well written, clearly visualised, the analyses are well conducted and (mostly) correctly interpreted. I have a few (major) remarks on the AMOC stability indicator (the $F_{ovS}$) and the link with an increased AL salinity transport. The comments below need to be addressed before I recommend the manuscript for publication.

We would like to thank the reviewer for his time reviewing the manuscript and for providing constructive comments and suggestions. The reviewer's expertise and insights on AMOC stability are very appreciated. Thank you for the thorough explanations of involved concepts!

Major comments and suggestions:

1. The parts of the manuscript which discuss the $F_{ovS}$ changes from AL and links with AMOC stability need to be more carefully stated. The interpretation is not always correct and a few arguments are missing, see the following points:

   • The AMOC carries relatively salty water northward in the North Atlantic Ocean, the local $F_{ov}$ is negative (e.g., at 40°N, see Jüling et al., 2021). When a freshwater perturbation is applied in the North Atlantic Ocean (in a hosing set-up), the AMOC strength and associated salinity transport reduce. The reduced salinity transport may amplify the original freshwater perturbation, leading to an even greater freshwater perturbation and further decreasing the AMOC strength: the (positive) salt-advection feedback. This feedback is only effective (see section 4b in Huisman et al., 2010) when velocity-induced and salinity-induced freshwater transport changes (under a freshwater perturbation) do not oppose each other. This is only the case when the AMOC carries net salinity into (exports net fresh water out of) the Atlantic basin, and hence $F_{ovS} < 0$. For the case when $F_{ovS} > 0$, the North Atlantic freshwater perturbations are usually 'flushed out' of the Atlantic Ocean and there is no positive salt-advection feedback. So the quantity $F_{ovS}$ only represents whether the AMOC amplifies (North Atlantic) freshwater perturbations, this is mentioned by the authors (line 353).

   The study by Haines et al. (2022) questions whether the $F_{ovS}$ is a useful metric for AMOC stability analysis in fully-coupled climate models (under constant pre-industrial conditions). They show that $F_{ovS}$ changes hardly influence the North Atlantic freshwater transport and a North Atlantic freshwater change is needed to modify the AMOC strength (Rahmstorf, 1996). However, van Westen et al. (2024a) demonstrated that the $F_{ovS}$ is a useful metric for AMOC stability analysis in the (low-resolution) CESM and this was consistent with previous work (e.g., Huisman et al., 2010). The differences between Haines et al. (2022) and van Westen et al. (2024a) could be related to the magnitude of the freshwater perturbations, where the latter study varies a North Atlantic freshwater flux forcing between 0 and 0.66 Sv.

   Relatively small freshwater/salinity perturbations at 34°S may be ineffective in modifying the North Atlantic Ocean freshwater content. This doesn't imply that there are no relations between

AMOC strength and $F_{ovS}$ (e.g., Figure 8a in van Westen and Dijkstra, 2024). The $F_{ovS}$ is also positive in the CESM (before 2070, Figure 10) and in this regime it is not very likely that AL changes destabilise the AMOC. The authors could argue that a greater AL salinity transport under climate change is preconditioning the AMOC to a more sensitive regime. You could also use the arguments that the CESM has known freshwater transport biases (van Westen and Dijkstra, 2024) and the observed $F_{ovS}$ is negative (Arumı́-Planas et al., 2024). My main point here is that the conclusions drawn from the AL changes on AMOC stability (e.g., lines 468 – 469) are sometimes strongly phrased. These parts need to be revised and a better discussion on the role of $F_{ovS}$ is needed (in both the introduction and discussion).

- To continue with my previous point: the interpretation of $F_{ovS}$ changes under strong transient responses. The quantity $F_{ovS}$ can only be used under (quasi-)equilibrium conditions (Rahmstorf, 1996) and these conditions include the equilibration of the Atlantic Ocean interior. Under strong climate change (RCP8.5) or large freshwater flux changes (Orı́huela-Pinto et al., 2022; Jackson et al., 2022) the $F_{ovS}$ responses and its effect on AMOC stability are much more difficult to interpret. It should be noted that the AMOC responses under climate change are dependent on the initial/historical $F_{ovS}$ value (Liu et al., 2017; van Westen and Dijkstra, 2024). The AL influences the $F_{ovS}$ under climate change (Figure 10) and directly connecting this to AMOC stability is difficult, as we consider the transient case. Once the CESM is equilibrated to the new 2100 radiative forcing conditions, the more negative $F_{ovS}$ suggests that the AMOC is closer to its tipping point, but this can't be verified from the transient results. Again, a more careful interpretation is needed when analysing the transient $F_{ovS}$ and AMOC responses.

- Lines 66 and 357: The authors suggest that a negative $F_{ovS}$ (in the AMOC on state) allows for a bi-stable AMOC regime. This is indeed the case in many conceptual (AMOC) models and models of intermediate complexity (Dijkstra 2024). However, the recent quasi-equilibrium hysteresis hosing simulation performed with a low-resolution version of the CESM (van Westen and Dijkstra, 2023) indicates that positive $F_{ovS}$ values (in the AMOC on state) are also part of the bi-stable AMOC regime. Sea-ice feedbacks, which were poorly captured in idealised models, modify the AMOC hysteresis behaviour (van Westen et al., 2024b) and allow for positive $F_{ovS}$ values to be part of the bi-stable AMOC regime. Climate model biases also shift the saddle-node bifurcations and negative $F_{ovS}$ does not exclusively indicate the bi-stable AMOC regime (Dijkstra and van Westen, 2024). Recent work by Lohmann et al. (2024) demonstrated a multi-stable AMOC regime under varying freshwater flux forcing, so not only bi-stable.

We thank the reviewer for clarifying the role of Fov in AMOC stability! Our statements were indeed too strong and we now just suggest that there can be a connection between the AL salt transport and AMOC stability in a warming climate. We have changed the introduction part about this to include the more current research and discussed our findings with respect to these and added all the difficulties of drawing this connection. Furthermore, we have changed our conclusions and now just state that there can be an influence while highlighting the need for further research here.

Abstract, l.16:
The increase in Agulhas leakage is accompanied by a higher salt transport into the Atlantic Ocean, which could play a role in the stability of the AMOC by the salt-advection-feedback.

Introduction, l.67ff:
A negative freshwater transport describes a bi- or multi-stable AMOC where a sudden shift in the freshwater forcing can lead to an AMOC collapse (Rahmstorf, 1996; Westen et al., 2024b; Lohmann et al., 2024). Observations of the real ocean estimate a negative freshwater transport (Arumí-Planas et al., 2024). Climate models exhibit the full range of values while a positive value can also be part of a bi-stable AMOC regime in some models (Westen et al., 2024a). The impact of the Agulhas leakage on this freshwater transport and further impacts on the AMOC remain to be completely understood (Weijer et al., 2019).

Results, l,371:
Removed: "However, quantifying the impact of the Agulhas Leakage change ultimately on the AMOC is not straightforward, because so many other factors have an influence on AMOC. The freshwater transport reaches negative values towards the end of the 21st century, which then implies a positive salt-advection feedback and an even stronger AMOC decrease in a bi-stable regime."

Discussion,
l.435:
"In this study we were able to identify a strong negative correlation between the salt transport and the freshwater transport across 34°S.  Even though it is likely that changes in Agulhas leakage will have an impact on the stability of the AMOC, the strong trend of Fov , including a sign change (Figure 10), does not allow a direct conclusion. Dedicated studies are required, optimally with sensitivity experiments using coupled models (e.g., Schulzki et al., 2024)."

l.464ff:
"The increased salt transport then contributes to a decrease of the freshwater transport. Westen and Dijkstra (2023) also show that the decrease in the freshwater transport is salinity based and mostly depends on changes in the upper 1500m, which is consistent with the impact of the Agulhas leakage salt transport. However, the freshwater transport as a metric for AMOC stability only really holds under equilibrium conditions (Rahmstorf, 1996). Additionally, the change of the AMOC due to global warming also strongly depends on the initial AMOC state (Liu et al., 2017). As our simulations do not reach an equilibrium by 2100, the discussion on a potential connection between Agulhas leakage and the stability of the AMOC has to be considered with care and would require a set of sensitivity experiments that is beyond this study. The hypothesis remains that the salt input from the Agulhas leakage into the Atlantic ultimately reaches the North Atlantic and deep water formation regions (Weijer and van Sebille, 2014). The salt can then play a role in setting the local stratification and thereby positively impacting deep water formation north of south the Greenland-Scotland Ridge. This has been described in coupled model experiments by Schulzki et al. (2024). However, owing to the long timescales involved, the direct quantification of these processes and the question on the stability of the AMOC strength needs further research. For this purpose, one would need ensemble experiments with eddy-rich Atlantic-wide or global configurations and long experiments with predicted or idealised freshwater hosing."

Conclusion, l.490:
"Nevertheless, the moderate Agulhas leakage increase under a

warming climate is accompanied by an increasing salt transport that could play a role in the stability of the AMOC."
Removed: "The reduced stability combined with the weakening of the strength of AMOC under a warming climate exacerbate the potential of a future AMOC collapse."

2. Line 227: The total freshwater transport at 34°S can be decomposed (Jüling et al., 2021) into four different contributions: overturning, azonal (gyre), barotropic (≈ 0) and eddies. I would argue that Agulhas rings, which are part of the AL, would end up in the eddy component and not (directly) in the overturning component. Have you considered determining the eddy-induced freshwater transport by AL? In Jüling et al. (2021) there is a negative freshwater transport trend in the eddy component (their Figure 7). I agree with the authors that oceanic adjustment by Agulhas rings or Rossby waves can eventually influence the overturning component (line 297), but this is relevant on time scales longer than one year.

Thank you for these points! We calculated the eddy-component but there was no correlation between AL salt flux and F_eddy. The meridional salt transport from the model, which is needed for the calculation of Feddy, was also only available for 220 years, so 150 years less than the rest of the data, which influences the results of the correlation. F_eddy might also be influenced by other variabilities along the section like eddies in the Brazil/Malvinas-current. We have added a couple sentences in the results section as follows:

L.323:
"In addition to the overturning component, the meridional freshwater transport in the Atlantic Ocean has an eddy component (Jüling et al., 2021). One could expect that the chaotic nature of Agulhas rings could lead to a connection here. However, we do not find a significant correlation between the eddy component of the freshwater transport and the Agulhas leakage salt transport. A caveat here is that , the available time series necessary for the calculation was 150 years shorter than for the other data which influences the results,questioning its robustness."

The time scale of one year in line 297 is for the connection between the Agulhas Current and the Agulhas Leakage, blue line in

Fig. 6. This is the time for the particles to be advected from the Agulhas Current to the Good Hope section where Agulhas Leakage is calculated. The time scale for the connection of Agulhas Leakage to the overturning component is around three years, orange and green lines in Fig. 6, which fits to ring or adjustment processes.

3. A low-resolution (1°) companion CESM simulation is available within the iHESP project (line 120). Climate model projections at 34°S (and elsewhere) are model resolution dependent (van Westen and Dijkstra, 2024) and could be relevant in the AL responses. Such a resolution comparison is also useful when considering other (1° resolution) CMIP6 models. I was wondering why the low-resolution CESM was not included in the analysis, at least a clear motivation is missing.

I would like to encourage the authors to conduct the analysis on the most interesting quantities in the low-resolution version of the CESM from the iHESP project. For example, it would be very interesting to see Figures 9 and 10 for the low-resolution CESM and to compare against the high-resolution CESM results. The low-resolution CESM results should go in the Appendix and are discussed in the main text. I'm not asking for a complete low-resolution CESM nor CMIP6 analysis, this is too much work and beyond the scope of the paper. The high-resolution CESM results are (and should be) central here.

A comparison of our results with simulations with lower resolution models is certainly illustrative. However, since we already know that the exact amount and timing of Agulhas rings are largely dependent on the dynamics of the Agulhas Current, its retroflection, and ring shedding, and therefore requires simulations at eddy-rich resolution. Therefore, we do not think that an expansion of the manuscript will be a useful exercise.

Minor comments and suggestions:

1. Lines 1, 20, 25, and throughout manuscript: 'the warm and salty waters'. Refer to the *relatively* warm and salty waters (or quantify warm and salty waters). Please check and fix throughout manuscript.

Changed these occurrences throughout.

2. Line 41: Maybe helpful for the reader to provide the time-mean formation rates and propagation speeds of the Agulhas 'eddies'.

Following the reviewer's suggestion, we have added this information on lines 42 as follows: "They have an average radius of

150 − 200 km, about 5 - 6 rings form per year and then propagate into the Atlantic at a speed of 5 − 15 km/day (Schouten et al., 2000)."

3. Throughout manuscript: Perhaps use Agulhas *rings* instead of Agulhas *eddies*. Ocean eddies arise from baroclinic instabilities, while ring shedding is a different processes.

   You are correct. All occurrences have been modified accordingly.

4. Line 58: The salt-advection feedback is the dominant destabilising AMOC mechanism and has been demonstrated in climate/AMOC models of varying complexity (Dijkstra, 2024). This sentence suggests that there are more (equally important?) feedback mechanisms, which ones did you consider? Note that the AMOC can be weakened through rapid climate change (e.g., G´erard and Crucifix, 2024) or large freshwater flux change (Or´ıhuela-Pinto et al., 2022), but these AMOC responses are related to the imposed forcing and not to a self-amplifying feedback loop (see also Major point 1).

   You are right that this is the dominant mechanism. We changed it to "A major theory involved here is the salt-advection feedback." l. 60. Still, this does not state (or imply) that any other related mechanisms have to be necessarily a feedback. Rather they can be anything, including, e.g., freshwater flux changes as mentioned by the reviewer.

5. Line 67: '... lead to an AMOC collapse (Rahmstorf, 1996)'. I would add the study by van Westen et al., (2024a), they demonstrate an AMOC collapse in a modern complex climate model under quasi-equilibrium hosing conditions. The study by Dijkstra (2024) is also relevant here, as it provides a review of AMOC tipping behaviour in a hierarchy of climate models.

   Following the reviewer's suggestion, we have added the following sentence: "

   "A negative freshwater transport describes a bi- or multi-stable AMOC where a sudden shift in the freshwater forcing can lead to an AMOC collapse (Rahmstorf, 1996; Westen et al., 2024b; Lohmann et al., 2024)."

6. Line 97 – 98: Only one RCP scenario is available within the iHESP project, namely the RCP8.5 scenario. This suggests that more RCP scenarios are available for the CESM and you only selected the RCP8.5 for the analysis. Please refer to the RCP8.5 scenario here (or motivate why you have focussed on the RCP8.5 scenario).

We have adapted the sentence: "The available high-resolution CESM simulations include a 500-year pre-industrial control (PIcontrol) run and a 3-member ensemble of historical and transient simulations in which the transient (future projection) period uses the Representative Concentration Pathway 8.5 (RCP8.5)." l. 99

7. Line 120: 'lower resolution counterpart' → lower (1°) resolution counterpart. Good to quantify the resolution here.

Agreed. We have added the following sentence:

"It has been shown that this simulation captures many features of the current climate well, showing improvements compared to a lower resolution counterpart (1°) (Chang et al., 2020).

8. Lines 127, 151, 161, etc.: 'absolute dynamic topography (Sea Surface Height above geoid)'. I would recommend to use 'dynamic sea level (DSL)', following the terminology proposed by Gregory et al. (2020). The DSL corresponds to the 'SSH' variable from the CESM output.

Thank you for pointing that out. We have changed it throughout the manuscript.

9. Line 128: 'model's capacity', which model are you referring to? The FOSI?

This paragraph is just about introducing the SSH satellite data. So, we have not modified the paragraph. However, we have more specifically described the simulation used for the comparison in the analysis section. Specifically, we use the CESM fully-coupled simulations for comparison, not FOSI.

10. Line 129: This product → The altimetry product. The reference to 'this product' is not clear to me.

Changed 'product' to 'data', l.130

These data are provided by the Copernicus Marine Environment Monitoring Service (CMEMS) on a global grid of 0.25° resolution and are based on a data unification and altimeter combination system (Mertz et al., 2017).

11. Line 207: Figure A5, maybe re-order the appendix figures so that they appear in their reference order as in the main text. So Figure

A5 → Figure A2 (and move/re-label these figures in the Appendix).

Thank you for this suggestion. The figures have been reordered..

12. Line 226: salt flux and freshwater flux. I suggest to use salt *transport* and freshwater *transport,* which is then consistent with the definition used in line 227 (annual freshwater transport). Check this throughout the manuscript.

Done.

13. Line 244: Upsream → Upstream (typo)

Corrected.

14. Line 247: In the model → In the CESM (mention the CESM here).

Incorporated as follows l.247:

"To determine the fidelity of our CESM simulations in resolving the necessary dynamics in our region of interest, we compare the Dynamic Sea Level variability from the PIcontrol simulation to that of satellite observations in Figure 3.

15. Line 264: Is this the variability between 2 – 5 years, the period with significant (95%-CI) peaks? Good to quantify the 'inter-annual variability' here.

Clarified as follows , l.270:

"The spectrum of the annual-mean timeseries presented in Figure 4b shows that the interannual variability with a period of two years clearly stands out, which is related to the formation and propagation of Agulhas Rings (Holton et al., 2017)."

16. Line 265: Alternatively, you could determine the confidence level at which the 40–50 year period is significant (similar to the p-value of 0.06, line 169).

We prefer to just say that it is insignificant, because adding a specific significance level does not add more.

17. Line 405: transport volume → volume transport

Done.

18. Line 410: A recent study by Arumˊı-Planas et al. (2024) quantified the $F_{ovS}$ from available observations, the present-day $F_{ovS}$ is indeed negative. This study is worth mentioning here.

Thanks for bringing this manuscript to our attention. It has been added to both the introduction and the discussion.

References:

1. Arumˊı-Planas et al., (2024): A Multi-Data Set Analysis of the Freshwater Transport by the Atlantic Meridional Overturning Circulation at Nominally 34.5S, https://doi.org/10.1029/2023JC020558

2. Dijkstra (2024): The role of conceptual models in climate research, https://www.sciencedirect.com/science/article/pii/S0167278923000338X

3. Dijkstra and van Westen (2024): The Effect of Indian Ocean Surface Freshwater Flux Biases On the Multi-Stable Regime of the AMOC, https://a.tellusjournals.se/articles/10.16993/tellusa.3246

4. Jackson et al., (2022): Understanding AMOC stability: the North Atlantic Hosing Model Intercomparison Project, https://doi.org/10.5194/gmd16-1975-2023

5. Gˊerard and Crucifix (2024): Diagnosing the causes of AMOC slowdown in a coupled model: a cautionary tale, https://doi.org/10.5194/esd-15293-2024

6. Gregory et al., (2020): Concepts and Terminology for Sea Level: Mean, Variability and Change, Both Local and Global, https://link.springer.com/article/10.1007/s10712-019-09525-z

7. Haines et al., (2022): Variability and Feedbacks in the Atlantic Freshwater Budget of CMIP5 Models With Reference to Atlantic Meridional Overturning Circulation Stability, https://doi.org/10.3389/fmars.2022.830821

8. Huisman et al., (2010): An Indicator of the Multiple Equilibria Regime of the Atlantic Meridional Overturning Circulation, https://journals.ametsoc.org/view/journals/phoc/40/3/2009jpo4215.1.xml

9. Ju¨ling et al., (2021): The Atlantic's freshwater budget under climate

change in the Community Earth System Model with strongly eddying oceans, https://doi.org/10.5194/os-17-729-2021

10. Liu et al., (2017): Overlooked possibility of a collapsed Atlantic meridional overturning circulation in warming climate, https://www.science.org/doi/10.1126/sciadv.1601666

11. Lohmann et al., (2024): Multistability and intermediate tipping of the Atlantic Ocean circulation, https://www.science.org/doi/full/10.1126/sciadv.adi4253

12. Or´ıhuela-Pinto et al., (2022): Interbasin and interhemispheric impacts of a collapsed atlantic overturning circulation, https://doi.org/10.1038/s41558022-01380-y

13. Rahmstorf (1996): On the freshwater forcing and transport of the At-lantic thermohaline circulation, https://link.springer.com/article/10.1007/s003820050144

14. van Westen and Dijkstra (2023): Asymmetry of AMOC Hysteresis in a State-Of-The-Art Global Climate Model, https://doi.org/10.1029/2023GL106088

15. van Westen and Dijkstra (2024): Persistent climate model biases in the Atlantic Ocean's freshwater transport, https://doi.org/10.5194/os-20549-2024

16. van Westen et al., (2024a): Physics-based early warning signal shows that AMOC is on tipping course, https://www.science.org/doi/full/10.1126/sciadv.adk1189

17. van Westen et al., (2024b): The Role of Sea-ice Processes on the Probability of AMOC Transitions, https://arxiv.org/abs/2401.12615